# MODELLING BRAIN CONNECTOMES: SOLV IS A WORTHY COMPETITOR TO HYPERBOLIC GEOMETRY!

## ABSTRACT

Finding suitable embeddings for connectomes (spatially embedded complex networks that map neural conections in the brain) is crucial for analyzing and understanding cognitive processes. Recent studies has found two-dimensional hyperbolic embeddings superior to Euclidean embeddings in modelling connectomes across species, especially human connectomes. However, those studies had some limitations: geometries other than Euclidean, hyperbolic or spherical were not taken into account. Following the suggestion of William Thurston that the networks of neurons in the brain could be sucessfully represented in Solv geometry, we study goodness-of-fit of the embeddings for 21 connectome networks (8 species). To this end, we suggest an embedding algorithm based on Simulating Annealing that allows us embed connectomes to Euclidean, Spherical, Hyperbolic, Solv, Nil, and also product geometries. Our algorithm tends to find better embeddings than the state of the art, even in the hyperbolic case. Our findings suggest that while in many cases, three-dimensional hyperbolic embeddings yield the best results, Solv embeddings perform reasonably well.

## 1 INTRODUCTION

Connectomes are comprehensive maps of neural connections in the brain. Understanding the interactions shaped by them is a key to understanding cognititive processes. As connectomes are spatially embedded complex networks with the structure shaped by physical constraints and communication needs, they seem to be exhibit traits inherent to non-Euclidean geometries. That is why a vast amount of research interest has been recently devoted to finding the suitable embeddings for connectome networks. Recent studies (e.g., Whi et al. (2022); Allard & Serrano (2020)) have found two-dimensional hyperbolic embeddings superior to Euclidean embeddings in modelling connectomes across species, especially human connectomes. However, those studies had some limitations: geometries other than Euclidean, hyperbolic or spherical were not taken into account.

Our study broadens the perspectives for the suitable embeddings. We analyze the goodness-of-fit (measured with widely-used quality measures) of the embeddings for 21 connectome networks (8 species) to 15 unique tessellations (Euclidean, Spherical, Hyperbolic, Solv, Nil, and also product geometries). We include both two-dimensional manifolds and three-dimensional ones. Following the suggestion of William Thurston that the networks of neurons in the brain could be sucessfully represented in Solv geometry (one of eight so-called Thurston geometries), we stipulate that this using this geometry would outperform using hyperbolic geometry.

Against this background, our contribution in this paper can be summarized as follows:

- We provide a new embedding method based on Simulated Annealing (SA). Our experiments show that our algorithm tends to find better embeddings than the state of the art, even in the hyperbolic case, measured using the standard measures from the literature (mAP, MeanRank, greedy routing success and stretch).

- To our best knowledge, we are the first to compare embeddings of connectomes to all Thurston geometries. Thus, we introduce new possibilities in modelling of connectomes.

- We find that while in many cases three-dimensional hyperbolic geometry yields the best results, there are other geometries worth consideration, e.g., Solv. As our results are based on an extensive simulation scheme, they are more robust in comparison to previous work.

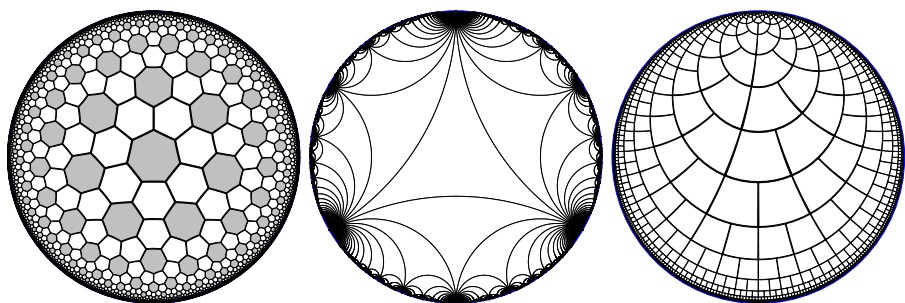

Figure 1: Tessellations of the hyperbolic plane. From left to right: (a) bitruncated order-3 heptagonal tiling ($\{7,3\}$), (b) infinite-order triangular tiling ($\{3,\infty\}$), (c) binary tiling.

## 2 HYPERBOLIC EMBEDDINGS

The $n$-dimensional sphere is $\mathbb{S}^n = \{x \in \mathbb{R}^{n+1} : g(x,x) = 1\}$, where $g$ is the Euclidean inner product, $g(x,y) = x_1y_1 + x_2y_2 + \ldots + x_{n+1}y_{n+1}$. The distance between two points $a$, $b$ on the sphere is the length of the arc connecting $a$ and $b$, which can be computed as $d(a,b) = \mathrm{acos}\, g(a,b)$. Similarly, $n$ dimensional hyperbolic geometry can be defined using the Minkowski hyperboloid model. In this model, $\mathbb{H}^n = \{x \in \mathbb{R}^{d+1} : x_{d+1} > 0, g^-(x,x) = -1$, where $g^-$ is the Minkowski inner product, $g^-(x,y) = x_1y_1 + x_2y_2 + \ldots + x_ny_n$. The distance is $d(a,b) = \mathrm{acosh}\, g^-(a,b)$.

Figure 1 depicts three tessellations of the hyperbolic plane $\mathbb{H}^2$ in the Poincaré disk model—a projection of $\mathbb{H}^2$ to the Euclidean plane that distorts the distances. In each of these tessellations, all the shapes (of the same color) are actually of the same hyperbolic size, even though ones closer to the boundary look smaller in the projection. Figure 1 shows the tree-like structure of hyperbolic geometry. This tree-likeness has found application in the visualization of hierarchical structures (Lamping et al., 1995; Munzner, 1998), and then in the modelling of complex networks. The hyperbolic random graph model (Boguñá et al., 2010) is parameterized by parameters $N$, $R$, $T$, $\alpha$. Each node $i \in \{1, \ldots, n\}$ is assigned a point $m(i)$ in the hyperbolic disk of radius $R$; the parameter $\alpha$ controls the distribution. Then, every pair of points $a, b \in \{1, \ldots, n\}$ is connected with probability $1/(1 + \exp((d - R)/T))$, where $d$ is the hyperbolic distance between $a$ and $b$. Graphs generated according to this model have properties typical to scale-free networks, such as high clustering coefficient and power law degree distribution (Papadopoulos et al., 2012; Boguñá et al., 2010).

## 3 THURSTON GEOMETRIES

By the uniformization theorem, every closed two-dimensional topological surface can be given spherical ($\mathbb{S}^2$), Euclidean ($\mathbb{E}^2$), or hyperbolic ($\mathbb{H}^2$) geometry, that is, there exists a Riemannian manifold with the same topology as $M$ and locally isometric to a sphere, Euclidean plane, or hyperbolic plane. William Thurston conjectured (Thurston, 1982) that three-dimensional topological manifolds can be similarly decomposed into fragments, each of which can be given one of eight *Thurston geometries*, which are homogeneous Riemannian manifolds. The eight Thurston geometries include:

- isotropic geometries: spherical ($\mathbb{S}^3$), Euclidean ($\mathbb{E}^3$), and hyperbolic ($\mathbb{H}^3$).
- product geometries: $\mathbb{S}^2 \times \mathbb{R}$ and $\mathbb{H}^2 \times \mathbb{R}$, In geometry $\mathbb{A} \times \mathbb{B}$, the distance $d_{\mathbb{A}\times\mathbb{B}}$ between $(a_1, b_1), (a_2, b_2) \in \mathbb{A} \times \mathbb{B}$ is defined using the Pythagorean formula:

$$d_{\mathbb{A}\times\mathbb{B}}((a_1, b_1), (a_2, b_2)) = \sqrt{d_{\mathbb{A}}(a_1, a_2)^2 + d_{\mathbb{B}}(b_1, b_2)^2}.$$

  Intuitively, the third dimension is added to $\mathbb{S}^2$ or $\mathbb{H}^2$ in the Euclidean way.
- Twisted product geometries: twisted $\mathbb{E}^2 \times \mathbb{R}$, also known as Nil, and twisted $\mathbb{H}^2 \times \mathbb{R}$, referred to as Twist in this paper, also known as the universal cover of $SL(2, \mathbb{R})$.
- Solv geometry, also known as Solve or Sol, which is fully anisotropic.

In low-dimensional topology, three-dimensional geometry is especially challenging, in particular, the Poincaré conjecture was the most challenging in three dimensions. On the other hand, our inter-

est in two-dimensional and three-dimensional geometries is based on their visualization possibilities (Kopczyński & Celińska-Kopczyńska, 2020; Coulon et al., 2020) and potential application to geometric embeddings. The original research into geometric embedding of networks used $\mathbb{H}^2$; more recently, higher-dimensional hyperbolic spaces are also studied (Jankowski et al., 2023; Whi et al., 2022). Similar embeddings are also used in machine learning, in particular, in Gu et al. (2019) product geometries are studied. Up to our knowledge, twisted product or Solv geometry have not been studied in this context. We are especially interested in the intriguing suggestion of William Thurston from 1997 that the architecture of brain might be based on Solv geometry (Schwartz, 2020).

The more exotic Thurston geometries have been successfully visualized only very recently (Kopczyński & Celińska-Kopczyńska, 2020; Coulon et al., 2020), and thus are much less known than isotropic geometries. We refer to these papers and explanatory videos (Rogue, 2023; 2022) and demos (Coulon et al., 2022) for detailed explanations of Solv and Nil geometries. In the rest of this section, we include a brief explanation of Solv and an intuitive explanation of twisted product geometries. We also discuss how their properties might prove beneficial for modeling networks.

To explain Solv, we should start first with the horocyclic coordinate system of $\mathbb{H}^2$. Horocycles are represented in the Poincaré disk model as circles tangent to the boundary; these can be seen as hyperbolic analogs of circles with infinite radius and circumference, centered in an ideal point (point on the boundary of the Poincaré disk). Concentric horocycles are seen in Figure 1c; the distance between two adjacent horocycles in this picture is $\log(2)$, and if two points $A$ and $B$ on given horocycle are in distance $x$, then the distance between their projections on the next (outer) horocycle is $2x$. For a point $P \in \mathbb{H}^2$, we project $P$ orthogonally to $Q$ on the horocycle going through the center $C$ of the Poincaré model. The $x$ coordinate is the (signed) length of the horocyclic arc $CQ$, and $y$ is the (signed) length of the segment $PQ$. (This is similar to the upper half-plane model (Cannon et al., 1997), except that we take the logarithm of the $y$ coordinate.) In this coordinate system, the length of the curve $((x(t), y(t)) : t \in [a,b])$ is defined as $\int_a^b \sqrt{(x'(t)\exp yt)^2 + y'(t)^2}dt$.

A similar coordinate system for $\mathbb{H}^3$ defines the length of the curve $((x(t), y(t), z(t)) : t \in [a,b])$ as $\int_a^b \sqrt{(x'(t)\exp z(t))^2 + (y'(t)\exp z(t))^2 + z'(t)^2}dt$. The surfaces of constant $z$ are called *horospheres*; the geometry on a horosphere is Euclidean. Solv geometry is obtained by switching the sign in this formula. That is, each point also has three coordinates ($x$, $y$ and $z$), but the length of a curve is now defined as $\int_a^b \sqrt{(x'(t)\exp z(t))^2 + (y'(t)\exp -z(t))^2 + z'(t)^2}dt$. The distance between two points in the length of the shortest curve connecting them; this length is difficult to compute (Coulon et al., 2020; Kopczyński & Celińska-Kopczyńska, 2022). Intuituively, the Solv geometry is based on two hierarchies (the hyperbolic plane $y =$const and the hyperbolic plane $x =$const), which are opposed to each other, due to the opposite sign used with $z$ in the distance formula. This gives us hope that Solv geometry can be used to represent hierarchies in three-dimensions which cannot be represented using other two- or three-dimensional geometries exhibiting simpler hierarchical structure ($\mathbb{H}^2$, $\mathbb{H}^3$, $\mathbb{H}^2 \times \mathbb{R}$). A similar effect of two opposing hierarchies could be also obtained in $\mathbb{H}^2 \times \mathbb{H}^2$, however, that is a four-dimensional geometry, and thus less suitable for visualization.

In Nil, we have well-defined directions in every point, which we can intuitively call North, East, South, West, Up and Down. However, while in Euclidean geometry, after moving 1 unit to the North, East, South, then West we return to the starting point, in Nil such a loop results in a move by 1 unit in the Up direction. In general, the vertical movement is equal to the signed area of the projection of the loop on the horizontal plane. Twist is based on the same idea, but the horizontal plane is now hyperbolic. An interesting property of Nil geometry is that it is a three-dimensional geometry where volume of a ball of radius $R$ has $\Theta(R^4)$ growth, which suggests better embedding possibilities than $\mathbb{E}^3$, but worse than the exponentially-expanding geometries.

## 4 OUR EMBEDDING ALGORITHM

Our goal is to find good quality embeddings of a connectome $(V, E)$ into some geometry $\mathbb{G}$, that is, a map $m : V \to \mathbb{G}$. As in the hyperbolic random graph model, we assume that our embedding has two parameters $R$ and $T$. The probability that an edge exists between $i$ and $j$ is again $p_1(d) = 1/(1 + \exp((d - R)/T))$, where $d$ is the distance between $m(i)$ and $m(j)$. We use MLE method to find the embedding, that is, we aim to maximize the likelihood $\prod_{1 \le i < j \le N} p(i,j)$, where $p(i,j) =$

$p_1(d_{\mathbb{G}}(m(i), m(j)))$ in case if the edge between $i$ and $j$ exists, and $p(i, j) = 1 - p_1(d_{\mathbb{G}}(m(i), m(j)))$ otherwise. Equivalently, we maximize the loglikelihood $\sum_{1 \leq i < j \leq N} \log p(i, j)$.

Prior algorithms learning embeddings may be specifically tailored to the specific geometry. Furthermore, prior algorithms assume that $d_{\mathbb{G}}$ is easy to compute, which is not the case for Solv. Therefore, a new embedding algorithm is necessary. As in Celińska-Kopczyńska & Kopczyński (2022), our algorithm is based on a uniform grid in geometry $\mathbb{G}$. Natural grids exist in all Thurston geometries of interest. While in the HRG model the network is mapped to a disk of radius $R$, here we map the network to the set $D$ of all grid points in $\mathbb{G}$ which are in distance at most $d_R$ from some fixed origin. We choose $d_R$ so that the number of points inside $D$ is fixed; in most experiments we pick $M = 20000$ points (actually, there may be slightly more points due to ties).

We compute the distance $d_{\mathbb{G}}$ for every pair of points in $D$, thus obtaining a $|D| \times |D|$ array that can be used to find the distance between pairs of points quickly. In case of Solv, it turns out that the method to compute the Solv distances from Kopczyński & Celińska-Kopczyńska (2020), while applicable to visualization, is not applicable to computing this table of distances due to long ranges. Therefore, for longer distances, we approximate by $d(a, b)$ as the smallest possible $d(a, a_1) + d(a_1, a_2) + \ldots + d(a_k, b)$, where intermediate points are also in $D$, and each pair of consecutive points is within the range of the exact method. Dijkstra's algorithm is used to find the path $(a_i)$.

Now, we use the Simulated Annealing (SA) method to learn the embedding. We start with an arbitrary embedding $m : V \to D$. Then, we perform the following for $i = 1, \ldots, N_S$. First, introduce a small change $m'$ to the current embedding $m$. Then, compute $L$, the loglikelihood of $m$, and $L'$, the loglikelihood of $m'$. If $L' > L$, always replace $m$ with $m'$. Otherwise, replace $m$ with $m'$ with probability $\exp((L' - L)/\exp(T))$, where the parameter $T$ depends on the iteration index.

In SA, we start with very high temperature $T$ (to accept all changes and thus explore the full space of possible embeddings without getting stuck on local maxima) and then we proceed to lower and lower temperatures (not accepting changes which yield much worse embeddings, but still experimenting with crossing lower valleys), eventually accepting only the changes which improve the embedding. In our experiments, $T$ decreases linearly from 10 to -15. We consider local changes of two possible forms: move $m'(i)$ for a random $i$ to a random point in $D$, and move $m'(i)$ for a random $i$ to a random point in $D$ that is close (neighbor) to $m(i)$.

We start with some initial values of $R$ and $T$. Occasionally during SA we find the values of $R$ and $T$ that best fit the current embedding, and we use the new values for the remaining iterations. Since finding the correct values takes time, we do it relatively rarely (every $|V|$ iterations with successful moves) and only once SA rejects most changes. In our experiments, we repeat this setup 30 times; in the following iterations, we start with the values of $R$ and $T$ of the best embedding found so far.

## 5   DATA, TESSELLATIONS, AND THE SETUP OF THE SIMULATION

Our implementation uses the tessellations implemented in RogueViz (Kopczyński & Celińska-Kopczyńska, 2023) and is based on the existing implementation of SA for finding hyperbolic visualizations (Celińska & Kopczyński, 2017). For our experiments, we use the same set of publicly available connectomes as Allard & Serrano (2020)[1]. See Table 1.

We run 30 iterations of SA to try to find the best $R$ and $T$, with $N_S = 10000 \cdot |V|$. In the literature, the quality of embeddings is usually evaluated using the *greedy routing* measures (in the network science community, Boguñá et al. (2010)) and MeanRank/mAP measures (in the machine learning community, Nickel & Kiela (2017)). Thus, we evaluate the quality of embeddings using the following five measures, from 0 (worst) to 1 (perfect).

SC Greedy routing success rate. This is the standard measure used in the literature on network embeddings (Boguñá et al., 2010). SC is the probability that, for random pair of vertices $(x, y) \in V^2$, the greedy routing algorithm starting at $x$ eventually successfully reaches the target $y$. This routing algorithm moves in the first step from $x$ to $x_1$, the neighbor of $x$ which is the closest to $y$ (that is, $d_{\mathbb{G}}(m(x_1), m(y))$ is the smallest). If $x_1 \neq y$, we continue to $x_2$, the neighbor of $x_1$ which is the closest to $y$, and so on.

---

[1]URL: `https://github.com/networkgeometry/navigable_brain_maps_data/`

| name | node | zone | $|V|$ | $|E|$ | source |
|------|------|------|------|------|--------|
| CElegans | cell | nervous system | 279 | 2290 | Varshney et al. (2011) |
| Cat1 | area | cortex | 65 | 730 | Scannell et al. (1995) |
| Cat2 | area | cortex and thalamus | 95 | 1170 | Scannell et al. (1999) |
| Cat3 | area | cortex | 52 | 515 | Scannell et al. (1999) |
| Drosophila1 | cell | optic medulla | 350 | 2886 | Shinomiya et al. (2022) |
| Drosophila2 | cell | optic medulla | 1770 | 8904 | Shinomiya et al. (2022) |
| Macaque1 | area | cortex | 94 | 1515 | Kaiser & Hilgetag (2006) |
| Macaque2 | area | cortex | 71 | 438 | Young (1993) |
| Macaque3 | area | cortex | 242 | 3054 | Harriger et al. (2012) |
| Macaque4 | area | cortex | 29 | 322 | Markov et al. (2013) |
| Mouse2 | cell | retina | 916 | 77584 | Helmstaedter et al. (2013) |
| Mouse3 | cell | retina | 1076 | 90810 | Helmstaedter et al. (2013) |
| Human1 | area | cortex | 493 | 7773 | Hagmann et al. (2008) |
| Human2 | area | cortex | 496 | 8037 | Hagmann et al. (2008) |
| Human6 | area | whole brain | 116 | 1164 | Gray Roncal et al. (2013) |
| Human7 | area | whole brain | 110 | 965 | Gray Roncal et al. (2013) |
| Human8 | area | whole brain | 246 | 11060 | Gray Roncal et al. (2013) |
| Rat1 | area | nervous system | 503 | 23029 | Bota & Swanson (2007) |
| Rat2 | area | nervous system | 502 | 24655 | Bota & Swanson (2007) |
| Rat3 | area | nervous system | 493 | 25978 | Bota & Swanson (2007) |
| ZebraFinch2 | cell | basal-ganglia (Area X) | 610 | 15342 | Dorkenwald et al. (2017) |

Table 1: Connectomes in our experiments. Based on Allard & Serrano (2020)

IST Greedy routing stretch. Stretch is the expected ratio of the length of the route found in the greedy routing procedure, to the length of the shortest route, under the condition that greedy routing was successful. IST is the reciprocal of stretch.

IMR For an edge $(x, y) \in E$, rank$(x, y)$ is 1 plus the number of vertices which are closer to $x$ than $y$, but not connected with an edge. MeanRank is the expected value of $(x, y)$ over all edges. We use IMR=1/MeanRank.

MAP For an edge $(x, y) \in E$, $P(x, y)$ is the ratio of vertices in distance of at most $d_{\mathbb{G}}(m(x), m(y))$ to $x$ which are connected with $x$. $AP(x)$ is the average of $P(x, y)$ for all $y$ connected with $x$, and MAP is the average of $AP(X)$ over all $X$ (MAP $\in [0, 1]$).

NLL Last but not least, loglikelihood (LL), which is directly maximized by us, as well as in many other embedding algorithms. For a given connectome $(V, E)$, the best theoretically possible loglikelihood is obtained when an edge between $x$ and $y$ occurs if and only iff the distance $d_{\mathbb{G}}(m(x), m(y))$ is below some threshold value and thus edges can be predicted with full certainty based on the distance (loglikelihood = 0), and the worst possible is obtained when the distance gives no information on edges, and thus the probability of each edge is predicted as $|E|/\binom{|V|}{2}$ (loglikelihood = H). Normalized loglikelihood, NLL, is defined as 1-LL/H, and is again from 0 to 1.

The computations of SC, STR, MR and MAP care on the order of nodes $y \in V$ by distance from $x \in V$. However, since we are using a discrete set $D$, it is possible that $d_{\mathbb{G}}(m(x), m(y)) = d_{\mathbb{G}}(m(x), m(z))$ for $y \neq z$. In the case of tie, we assume a random order of the tied nodes. During the statistical testing, where necessary, we apply Bonferroni correction for multiple testing.

In our main experiment, we work with the 15 unique tessellations listed in Table 2. Most of our tessellations are hyperbolic. Subdivided($d$) means that each cube of the honeycomb has been subdivided into $d \times d \times d$ subcubes, and the point $D$ consists of the vertices and centers of these subcubes, approximating the set of centers of cells of the Euclidean bitruncated cubic honeycomb. In case of Nil and Solv, we do not get actual cubes, so this construction is approximate. For technical reasons, distances are rounded to the nearest integer multiple of 1/20 absolute unit, except sphere, where the unit is 1/200 of absolute unit. Thus, diameter 316 for a continuous tessellation is 15.8 absolute units, and sphere has diameter (i.e., half the circumference) $\pi$.

| name | dim | geometry | closed | nodes | diameter | description of the set $D$ |
|---|---|---|---|---|---|---|
| $\mathbb{H}^2$ | 2 | hyperbolic | F | 20007 | 304 | bitruncated $\{7,3\}$ (Figure 1a) |
| $\mathbb{H}^2\&$ | 2 | hyperbolic | T | 17980 | 157 | closed hyperbolic manifold |
| tree | 2 | tree | F | 20002 | 396 | $\{3,\infty\}$ (Figure 1b) |
| $\mathbb{E}^3$ | 3 | euclid | F | 20107 | 1070 | bitruncated cubic honeycomb |
| $\mathbb{E}^3\&$ | 3 | euclid | T | 19683 | 450 | torus subdivided into $27 \times 27 \times 27$ cells |
| $\mathbb{H}^3$ | 3 | hyperbolic | F | 21365 | 201 | $\{4,3,5\}$ hyperbolic honeycomb |
| $\mathbb{H}^3*$ | 3 | hyperbolic | F | 20039 | 146 | $\{4,3,5\}$ subdivided(2) |
| $\mathbb{H}^3\&$ | 3 | hyperbolic | T | 9620 | 102 | subdivided(2) closed hyperbolic manifold |
| Nil | 3 | nil | F | 20009 | 1000 | $\mathbb{Z}^3$ grid |
| Nil* | 3 | nil | F | 20208 | 290 | $\mathbb{Z}^3$ grid, subdivided(2) |
| Twist | 3 | twist | F | 20138 | 152 | twisted $\{5,4\} \times \mathbb{Z}$ |
| $\mathbb{H}^2 \times \mathbb{R}$ | 3 | product | F | 20049 | 29 | bitruncated $\{7,3\} \times \mathbb{Z}$ |
| Solv | 3 | solv | F | 20017 | 246 | analog of Figure 1c |
| Solv* | 3 | solv | F | 20000 | 143 | analog of Figure 1c, subdivided(2) |
| $\mathbb{S}^3$ | 3 | sphere | T | 21384 | 628 | 8-cell, each cell subdivided(11) |

Table 2: Details on tessellations used in our study; * denotes finer grids.

## 6 COMPARISON AT MAXIMUM PERFORMANCES

We start with a naive comparison among the tessellations based on the best results that were obtained for each tessellation for each connectome. Due to space limitations, we have moved the ranking figures and descriptive statistics to Appendix D.

| connectome | NLL | MAP | IMR | SC | IST |
|---|---|---|---|---|---|
| Cat1 | 5.47 | 1.29 | 10.28 | 0.40 | 0.65 |
| Cat2 | 4.84 | 3.75 | 8.94 | 1.94 | 1.63 |
| Cat3 | 6.22 | 1.35 | 11.04 | 0.09 | 0.66 |
| CElegans | 7.46 | 6.05 | 8.38 | 8.89 | 6.30 |
| Drosophila1 | 5.46 | 10.15 | 8.34 | 12.19 | 9.47 |
| Drosophila2 | 12.52 | 32.87 | 11.48 | 27.32 | 25.87 |
| Human1 | 9.13 | 5.95 | 29.08 | 11.94 | 7.06 |
| Human2 | 9.19 | 6.20 | 28.38 | 11.62 | 7.00 |
| Human6 | 7.69 | 3.52 | 26.79 | 7.29 | 4.53 |
| Human7 | 8.13 | 3.45 | 25.58 | 7.23 | 4.34 |
| Human8 | 6.38 | 1.72 | 17.92 | 0.23 | 0.74 |
| Macaque1 | 3.95 | 3.93 | 10.21 | 2.87 | 2.21 |
| Macaque2 | 7.22 | 3.02 | 16.74 | 6.11 | 3.30 |
| Macaque3 | 4.99 | 7.52 | 9.05 | 6.88 | 5.84 |
| Macaque4 | 9.44 | 0.27 | 4.51 | 0.00 | 0.00 |
| Mouse2 | 9.68 | 7.54 | 10.86 | 3.78 | 4.94 |
| Mouse3 | 10.85 | 8.84 | 10.98 | 3.58 | 5.14 |
| Rat1 | 44.60 | 32.51 | 66.25 | 10.25 | 8.18 |
| Rat2 | 44.32 | 31.33 | 68.97 | 10.02 | 8.13 |
| Rat3 | 40.76 | 27.42 | 62.36 | 9.85 | 7.96 |
| ZebraFinch2 | 14.83 | 19.70 | 7.06 | 16.29 | 12.50 |

Table 3: Coefficients of variations (CV, in %) for the max performance of the geometries

According to Table 4, we notice that the assessment of the performance of the geometry may vary with respect to the quality measure; there are also differences across species. E.g., in general, trees perform poorly in terms of measures other than greedy success rate, and no matter the measure, they are always the best choice for Rat's connectomes (nervous system). Results for Rat's and Drosophila2's connectomes are also characterized by the relatively high variation among species (Table 3). For other species, the best performances are actually similar with respect to a quality measure: the differences in best performance among geometries measured with MAP, greedy rate success, and stretch are small (in most of the cases values of CVs are under 10%); especially for Cat's connectomes they tend to be negligible (values of CVs even under 1%).

| geometry | Top 5 ranks | | | | | Bottom 5 ranks | | | | |
|---|---|---|---|---|---|---|---|---|---|---|
| | NLL | MAP | IMR | SC | IST | NLL | MAP | IMR | SC | IST |
| $\mathbb{H}^2$ | 19.05 | 23.81 | 14.29 | 80.95 | 33.33 | 57.14 | 42.86 | 71.43 | 0.00 | 19.05 |
| $\mathbb{H}^2$& | 0.00 | 0.00 | 0.00 | 0.00 | 0.00 | 95.24 | 85.71 | 90.48 | 95.24 | 90.48 |
| tree | 23.81 | 23.81 | 14.29 | 80.95 | 47.62 | 66.67 | 42.86 | 80.95 | 0.00 | 28.57 |
| $\mathbb{E}^3$ | 19.05 | 23.81 | 23.81 | 9.52 | 14.29 | 57.14 | 66.67 | 38.10 | 61.90 | 57.14 |
| $\mathbb{E}^3$& | 19.05 | 28.57 | 47.62 | 0.00 | 4.76 | 52.38 | 57.14 | 33.33 | 90.48 | 85.71 |
| $\mathbb{H}^3$ | 66.67 | 61.90 | 33.33 | 52.38 | 66.67 | 9.52 | 14.29 | 42.86 | 0.00 | 0.00 |
| $\mathbb{H}^3$∗ | 66.67 | 76.19 | 38.10 | 61.90 | 76.19 | 0.00 | 0.00 | 4.76 | 0.00 | 0.00 |
| $\mathbb{H}^3$& | 9.52 | 19.05 | 28.57 | 0.00 | 4.76 | 14.29 | 14.29 | 4.76 | 90.48 | 66.67 |
| Nil | 19.05 | 9.52 | 33.33 | 4.76 | 0.00 | 4.76 | 9.52 | 4.76 | 4.76 | 9.52 |
| Nil* | 38.10 | 38.10 | 57.14 | 0.00 | 19.05 | 28.57 | 57.14 | 19.05 | 14.29 | 28.57 |
| Twist | 61.90 | 57.14 | 38.10 | 57.14 | 71.43 | 19.05 | 19.05 | 14.29 | 9.52 | 4.76 |
| $\mathbb{H}^2 \times \mathbb{R}$ | 66.67 | 52.38 | 52.38 | 42.86 | 71.43 | 0.00 | 0.00 | 0.00 | 0.00 | 0.00 |
| Solv | 52.38 | 47.62 | 33.33 | 47.62 | 42.86 | 14.29 | 14.29 | 28.57 | 9.52 | 4.76 |
| Solv* | 38.10 | 28.57 | 61.90 | 9.52 | 23.81 | 0.00 | 0.00 | 0.00 | 0.00 | 0.00 |
| $\mathbb{S}^3$ | 0.00 | 9.52 | 23.81 | 0.00 | 0.00 | 80.95 | 76.19 | 66.67 | 85.71 | 80.95 |

Table 4: Percentages: how often occurred within top or bottom five ranks (at the max performance)

The results suggest that $\mathbb{H}^2$& and $\mathbb{S}^3$ seem to be inefficient choices: the first one never enters the top five ranks; both often occur within the bottom five ranks, at their best performance being even the worst choices no matter the quality measure. In contrast, $\mathbb{H}^3$ and $\mathbb{H}^2 \times \mathbb{R}$ perform very well – they rarely occur within bottom five ranks. Twist and Solv or Solv∗ never happen to be the worst choices, all of them perform relatively well. Interestingly, the usage of finer grids may not increase the chance of obtaining the best performace, no matter the quality measure: while for $\mathbb{H}^3$∗ vs $\mathbb{H}^3$ and Solv* vs Solv we notice that it reduces the chance of occurring within the bottom five ranks, the best performances of non-fine grids still outperform them when it comes to the occurrences within the five top ranks. Contrary, finer grid for Nil significantly increases percentage of occurrences among five best ranks. When it comes to Euclidean geometry, the results are inconsistent. The best performances of $\mathbb{E}^3$ and $\mathbb{E}^3$& often occur among the bottom five ranks of the geometries. However, there are cases in which those geometries perform excellently, e.g., for Human connectomes.

## 7 COMPARISON OF PERFORMANCES BASED ON DISTRIBUTIONS

Comparison at the maximum performance from the previous section gives us intuition about the optimistic scenarios – what the limits for our embeddings are. However, due to the nature of SA, the maximum values we obtained are still realizations of random variables; that is why a closer inspection including information about the distributions of the simulation results is needed. To this end, we will compare geometries using voting rules, in particular, we will be interested in finding Condorcet winners and losers. As Condorcet winner may not exist in the presence of ties, we will refer to its simple modification: Copeland rule (Maskin & Dasgupta, 2004).

We say "geometry A wins against geometry B" if the probability that (for a given quality measure) a randomly chosen simulation result obtained by geometry A is greater than a randomly chosen simulation result obtained by geometry B is greater than 0.5. If that probability is equal to 0.5, we say that "there is a tie", and otherwise, "geometry A loses". To compute the score for a given geometry, we add 1 for every winning scenario, 0 for every tie, and -1 for every losing scenario. The geometries with the highest and lowest scores become Copeland winners and losers, respectively (we allow for more than just one candidate in both cases).

Condorcet winners (as well as the winners based on the Copeland method) have interpretations – those are the candidates that beat the most of other candidates in pairwise contests. In our case, we could perceive them as the best options for embeddings. Based on the data in Table 5, we cannot name one universal winner. While it seems that $\mathbb{H}^3$ is a sound choice, we also notice that Solv and Twist are worthy attention. Interestingly, for Human connectomes, $E^3$ outperforms other geometries. See Appendix C for weighted directed networks constructed upon the voting rules.

| | Copeland winners | | | | | Copeland losers | | | | |
|---|---|---|---|---|---|---|---|---|---|---|
| connectome | NLL | MAP | IMR | SC | IST | NLL | MAP | IMR | SC | IST |
| Cat1 | Solv* | $\mathbb{H}^3*$ | Solv* | $\mathbb{H}^3*$ | Solv* | $\mathbb{H}^2\&$ | tree | tree | $\mathbb{H}^2\&$ | tree |
| Cat2 | $\mathbb{H}^3*$ | $\mathbb{H}^3*$ | $\mathbb{H}^2\times\mathbb{R}$ | Twist | $\mathbb{H}^2\times\mathbb{R}$ | $\mathbb{H}^2\&$ | $\mathbb{S}^3$ | tree | $\mathbb{H}^3\&$ | tree |
| Cat3 | Solv* | Solv* | $\mathbb{H}^3\&$ | Nil* | $\mathbb{H}^3\&$ | $\mathbb{H}^2\&$ | tree | tree | $\mathbb{H}^2\&$ | tree |
| CElegans | $\mathbb{H}^3*$ | $\mathbb{H}^3$ | Nil | $\mathbb{H}^3*$ | Nil | $\mathbb{H}^2\&$ | $\mathbb{H}^2\&$ | tree | $\mathbb{H}^2\&$ | tree |
| Drosophila1 | Twist | $\mathbb{H}^3$ | $\mathbb{H}^3\&$ | $\mathbb{H}^3$ | $\mathbb{H}^3\&$ | $\mathbb{H}^2\&$ | $\mathbb{S}^3$ | tree | $\mathbb{H}^2\&$ | tree |
| Drosophila2 | $\mathbb{H}^3$ | $\mathbb{H}^3$ | $\mathbb{H}^3*$ | $\mathbb{H}^3$ | $\mathbb{H}^3*$ | $\mathbb{S}^3$ | $\mathbb{S}^3$ | $\mathbb{S}^3$ | $\mathbb{H}^3\&$ | $\mathbb{S}^3$ |
| Human1 | $\mathbb{E}^3$ | $\mathbb{S}^3$ | $\mathbb{S}^3$ | $\mathbb{H}^3*$ | $\mathbb{S}^3$ | tree | tree | tree | $\mathbb{H}^2\&$ | tree |
| Human2 | $\mathbb{E}^3$ | $\mathbb{S}^3$ | $\mathbb{S}^3$ | $\mathbb{H}^3*$ | $\mathbb{S}^3$ | tree | tree | tree | $\mathbb{H}^2\&$ | tree |
| Human6 | $\mathbb{E}^3$ | $\mathbb{E}^3$ | $\mathbb{E}^3$ | $\mathbb{H}^3*$ | $\mathbb{E}^3$ | tree | tree | tree | $\mathbb{H}^2\&$ | tree |
| Human7 | $\mathbb{E}^3$ | $\mathbb{E}^3$ | $\mathbb{E}^3$ | Solv | $\mathbb{E}^3$ | tree | tree | tree | $\mathbb{H}^2\&$ | tree |
| Human8 | $\mathbb{H}^3*$ | $\mathbb{H}^3*$ | $\mathbb{E}^3$ | $\mathbb{H}^2$ | $\mathbb{E}^3$ | tree | tree | tree | $\mathbb{H}^2\&$ | tree |
| Macaque1 | Solv | Solv | Solv | $\mathbb{H}^3*$ | Solv | $\mathbb{S}^3$ | $\mathbb{S}^3$ | tree | $\mathbb{E}^3\&$ | tree |
| Macaque2 | Nil | Nil | Nil* | $\mathbb{H}^2$ | Nil* | tree | tree | tree | $\mathbb{H}^2\&$ | tree |
| Macaque3 | $\mathbb{H}^3*$ | $\mathbb{H}^3*$ | $\mathbb{H}^2\times\mathbb{R}$ | $\mathbb{H}^2$ | $\mathbb{H}^2\times\mathbb{R}$ | $\mathbb{H}^2\&$ | $\mathbb{S}^3$ | tree | $\mathbb{H}^2\&$ | tree |
| Macaque4 | $\mathbb{E}^3\&$ | $\mathbb{E}^3\&$ | $\mathbb{E}^3\&$ | Twist | $\mathbb{E}^3\&$ | tree | tree | tree | $\mathbb{E}^3$ | tree |
| Mouse2 | Twist | $\mathbb{H}^3$ | $\mathbb{H}^2\times\mathbb{R}$ | $\mathbb{H}^2$ | $\mathbb{H}^2\times\mathbb{R}$ | $\mathbb{S}^3$ | $\mathbb{S}^3$ | $\mathbb{H}^2\&$ | $\mathbb{S}^3$ | $\mathbb{H}^2\&$ |
| Mouse3 | Twist | $\mathbb{H}^3$ | $\mathbb{H}^2\times\mathbb{R}$ | $\mathbb{H}^2$ | $\mathbb{H}^2\times\mathbb{R}$ | $\mathbb{S}^3$ | $\mathbb{S}^3$ | $\mathbb{S}^3$ | $\mathbb{H}^2\&$ | $\mathbb{S}^3$ |
| Rat1 | tree | tree | $\mathbb{H}^3$ | tree | $\mathbb{H}^3$ | $\mathbb{S}^3$ | $\mathbb{S}^3$ | $\mathbb{S}^3$ | $\mathbb{S}^3$ | $\mathbb{S}^3$ |
| Rat2 | tree | tree | $\mathbb{H}^3$ | tree | $\mathbb{H}^3$ | $\mathbb{S}^3$ | $\mathbb{S}^3$ | $\mathbb{S}^3$ | $\mathbb{E}^3\&$ | $\mathbb{S}^3$ |
| Rat3 | tree | tree | $\mathbb{H}^3$ | tree | $\mathbb{H}^3$ | $\mathbb{S}^3$ | $\mathbb{S}^3$ | $\mathbb{S}^3$ | $\mathbb{S}^3$ | $\mathbb{S}^3$ |
| ZebraFinch2 | Solv | $\mathbb{H}^3$ | Solv | $\mathbb{H}^3$ | Solv | $\mathbb{S}^3$ | $\mathbb{S}^3$ | $\mathbb{S}^3$ | Solv | $\mathbb{S}^3$ |

Table 5: Voting rules: Copeland winners and losers.

## 8 ROBUSTNESS CHECKS AND THREATS TO VALIDITY

Ideally, there exists optimal embedding of $(V, E)$ into the whole geometry $\mathbb{G}$, where $m_{\text{opt}} : V \to \mathbb{G}$, and some values of $R$ and $T$ are used. Unfortunately, the embedding $m$ found by SA might be worse than $m_{\text{opt}}$ due to the following issues. See Appendix B for a detailed analysis.

- The radius $d_R$ is too small, making $m_{\text{opt}}$ simply not fit,
- The grid used is too coarse, hence the necessity of making $m(i)$ the grid point to closest to $m_{\text{opt}}(i)$, and thus reducing the loglikelihood,
- The number of iterations of SA, $N_S$, is too small – while SA is theoretically guaranteed to find the optimal embedding for given $R$ and $T$ with high probability as $N_S$ tends to infinity, in practice, we are constrained by time limits,
- The values of the parameters $R$ and $T$ have not been chosen correctly.

**Our results vs previous approaches** To see how good is SA at obtaining good embeddings, we can compare it against the previously existing embedders. While we are the first to study Nil and Solv embeddings, there is a vast number of prior works on $\mathbb{H}^2$ and $\mathbb{H}^3$ embeddings. We have compared our results on the CElegans, Drosophila1, Human1 and Mouse3 connectomes. We use the results of comparison in Anonymous (2023). For $\mathbb{H}^2$, we have compared against the BFKL embedder (Bläsius et al., 2016), Mercator (García-Pérez et al., 2019) (fast and full version), 2D Poincaré embeddings (Nickel & Kiela, 2017) and 2D Lorentz embeddings (Nickel & Kiela, 2018). Each of the competing algorithms has been run five times, found the best result of these 25 runs, and compared to our results. We have also performed a similar analysis for $\mathbb{H}^3*$, against 3D Poincaré embeddings (BFKL and Mercator work only in $\mathbb{H}^2$). Table 6 list our results for mAP and success rate (see Appendix E for other measures).

In most cases, our result turned out to give better result in all 30 runs, and in almost all cases, we have received better results in most of the runs. We have not managed to beat Poincaré 3D embeddings on greedy success ratio and greedy stretch measures for Mouse3 and CElegans. Furthermore, our embeddings use smaller radius (7.7 for $\mathbb{H}^2$, 3.7 for $\mathbb{H}^3$), and use less time than Lorentz or Poincaré embeddings (about 220 seconds per run on Mouse3 in $\mathbb{H}^3$). Smaller radius means that our em-

| connectome | dim | mAP | method | rad | time | ours | better |
|---|---|---|---|---|---|---|---|
| celegans | 2 | 0.500 | Poincaré | 7.2 | 278 | 0.540 | 30 |
| celegans | 3 | 0.583 | Poincaré | 10.1 | 274 | 0.584 | 21 |
| drosophila1 | 2 | 0.425 | Mercator (full) | 23.6 | 14 | 0.483 | 30 |
| drosophila1 | 3 | 0.488 | Poincaré | 11.4 | 365 | 0.512 | 30 |
| human1 | 2 | 0.651 | Lorentz | 10.8 | 1085 | 0.675 | 30 |
| human1 | 3 | 0.722 | Poincaré | 9.4 | 827 | 0.799 | 30 |
| mouse3 | 2 | 0.585 | Mercator (full) | 29.9 | 117 | 0.612 | 30 |
| mouse3 | 3 | 0.654 | Poincaré | 12.2 | 9207 | 0.655 | 18 |
| connectome | dim | success | method | rad | time | ours | better |
| celegans | 2 | 0.903 | Poincaré | 7.2 | 267 | 0.931 | 27 |
| celegans | 3 | 0.958 | Poincaré | 10.1 | 274 | 0.930 | 0 |
| drosophila1 | 2 | 0.769 | Mercator (full) | 23.6 | 14 | 0.847 | 30 |
| drosophila1 | 3 | 0.844 | Poincaré | 11.4 | 365 | 0.843 | 13 |
| human1 | 2 | 0.889 | Poincaré | 12.2 | 1185 | 0.929 | 21 |
| human1 | 3 | 0.926 | Poincaré | 9.5 | 835 | 0.958 | 24 |
| mouse3 | 2 | 0.962 | Mercator (full) | 34.5 | 74 | 0.967 | 30 |
| mouse3 | 3 | 0.971 | Poincaré | 12.2 | 8679 | 0.952 | 0 |

Table 6: Our embeddings versus state-of-the-art. For each connectome and dimension, we list the best prior method and its result, the radius of the embedding, time elapsed in seconds, the best result of our method, and how many times (out of 30) our result was better.

beddings avoid numerical precision issues that tend to be a serious issue in hyperbolic embeddings (Bläsius et al., 2018; Sala et al., 2018; Celińska-Kopczyńska & Kopczyński, 2022), are better able to fully use both the larg-scale (tree-like) and smaller-scale (Euclidean-like) nature of hyperbolic geometry (while large radius embeddings tend to be tree-like), and making them more applicable for visualization (in large-radius visualizations, less nodes are visible).

## 9 CONCLUSIONS

In this paper, we presented an experimental analysis of embeddings of 21 connectomes to various geometries (both three- and two-dimensional). To our best knowledge, we are the first to compare embeddings to all Thurston geometries. We provided a new embedding method based on Simulated Annealing (SA) that outperforms previous methods.

Although earlier studies suggested one universal winner geometry (usually pointing at $\mathbb{H}^2$), our results showed that if we allow for the third dimension, the universal winner ceases to exists. Especially, $\mathbb{H}^2$ embeddings tend to be worse than (non-Euclidean) 3D geometries, even if our $\mathbb{H}^2$ embeddings are actually good – better than Bläsius et al. (2016); García-Pérez et al. (2019); Nickel & Kiela (2017; 2018). If we were to suggest a set of geometries that are worth attention while modelling connectomes, we would name $\mathbb{H}^3$, Solv, Twist, and $\mathbb{H}^2 \times \mathbb{R}$. Surprisingly, for Human connectomes, $\mathbb{E}^3$ is a suitable choice. There might be a correlation between the zone of the connectome and the best choice for the embedding, e.g., trees model nervous systems well.

Our results were based on an extensive simulation scheme with numerous robustness checks. While our results regarding log-likelihood, MAP, and MeanRank were similar and robust to the changes in the setup of SA, we noticed that optimizing log-likelihood may affect the quality measured by greedy success rate and stretch. We suppose that an explanation lies in capturing different aspects (functions) of the networks by those two groups of quality measure. Finding out the relationships among connectomes or embeddings characteristics and quality measures exceeds the scope of this paper and will be a subject of a future work.

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

## A   OTHER METHODS OF MEASURING DISTANCES

To pick the method of measuring the distances in our experiment, we start with the preliminary list of tessellations shown in Table 7. In the tessellations marked with (d), distances are computed as the lengths of the shortest paths in the graph $(D, E_D)$ where two points in $D$ are connected when they correspond to adjacent tiles of the tessellation (to make the distances comparable, the diameters in Table 7 are again multiplied by 20), while in the tessellations marked with (c), distances are computed according to the underlying geometry. In each pair, the sets $D$ used are roughly of the same size (we have less control over $|D|$ in discrete tessellations). In case of the product geometry

| name | dim | geometry | closed | nodes | diameter | description of the set $D$ |
|---|---|---|---|---|---|---|
| $\mathbb{H}^2$ (d) | 2 | hyperbolic | F | 27000 | 560 | bitruncated $\{7,3\}$ (Figure 1a) |
| $\mathbb{H}^2$ (c) | 2 | hyperbolic | F | 27007 | 316 | bitruncated $\{7,3\}$ (Figure 1a) |
| tree (c) | 2 | tree | F | 20002 | 396 | $\{3,\infty\}$ (Figure 1b) |
| tree (d) | 2 | tree | F | 24574 | 520 | binary tree |
| $\mathbb{H}^3$ (c) | 3 | hyperbolic | F | 40979 | 214 | $\{4,3,5\}$ hyperbolic honeycomb |
| $\mathbb{H}^3$ (d) | 3 | hyperbolic | F | 41511 | 280 | $\{4,3,5\}$ hyperbolic honeycomb |
| $\mathbb{H}^2 \times \mathbb{R}$ (c) | 3 | product | F | 20049 | 222 | bitruncated $\{7,3\}$) times $\mathbb{Z}$ |
| $\mathbb{H}^2 \times \mathbb{R}$ (a) | 3 | product | F | 20022 | 5637 | bitruncated $\{7,3\}$) times $\mathbb{Z}$ |

Table 7: Details on the preliminary tessellations used in our study.

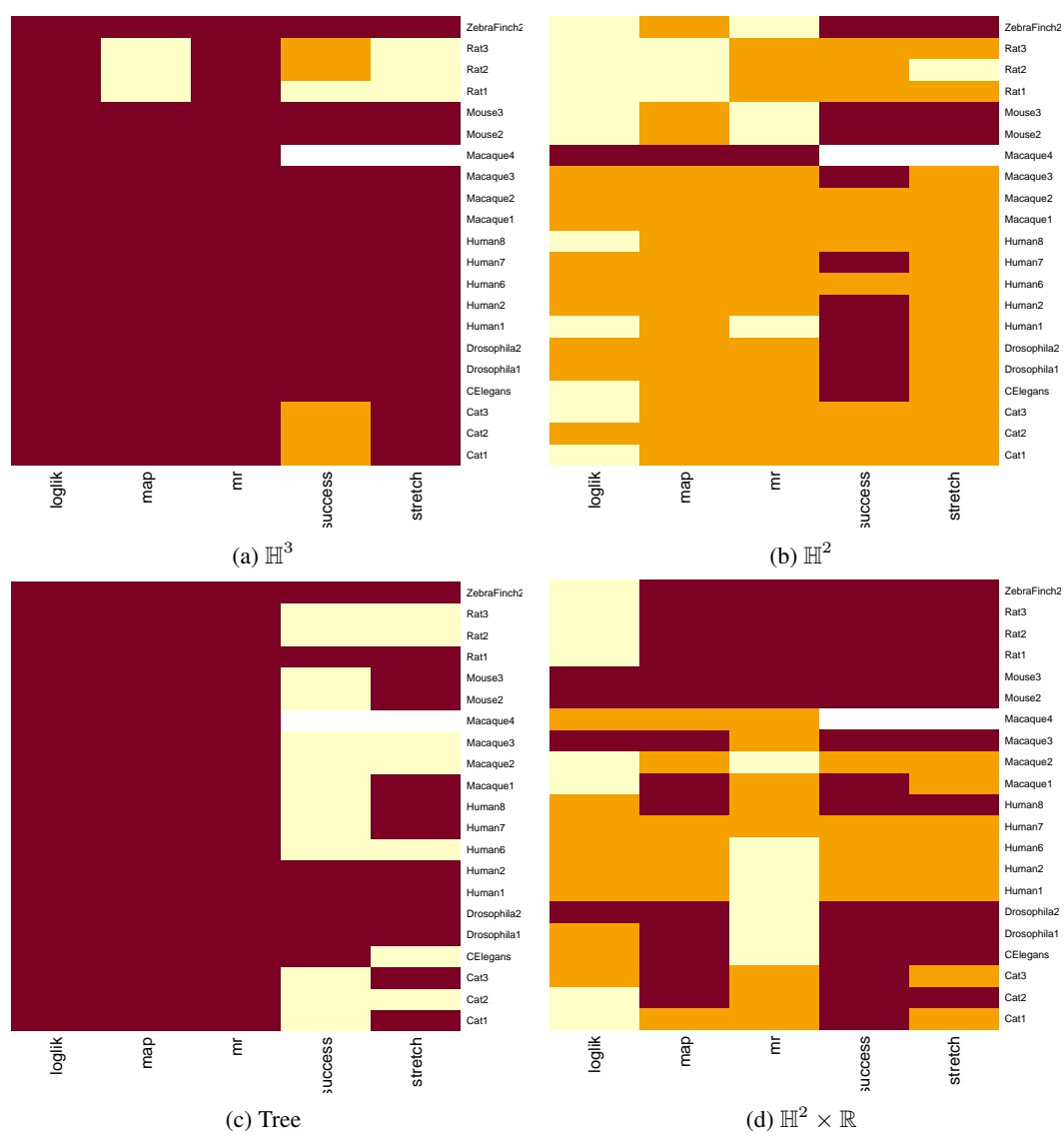

Figure 2: Comparison of the goodness-of-fit between pairs of tessellations. Red suggests that the continuous (non-angular) version yields better results and the difference is significant; orange suggests lack of significant difference, and yellow suggests significantly worse results for continuous (non-angular) version, respectively.

$\mathbb{H}^2 \times \mathbb{R}$, we compare geometric distances (c) to angular distance (a). The angular distance $d_a(X,Y)$

is, intuitively, how small an object at $Y$ appears to an observer placed at $X$, assuming that the light travels along geodesics. The angular size of an object in distance $d$ is proportional to $1/d$ in the Euclidean case and $1/\exp(d)$ in the hyperbolic case; for anisotropic geometries, it may depend on the axis. More precisely, $d_a(X, Y)$ is proportional to $\lim_{r \to 0} r^2/p(X, Y, r)$, where $p(X, Y, r)$ is the probability that a random geodesic starting in $X$ passes within distance of at least $r$ from $Y$.

It was not certain if we benefit from those technical subtelties. As the data is not normally distributed and the sample sizes are small (30 observations), we perfomed Wilcoxon tests (with Bonferroni correction for multiple testing). Figure 2 visualizes the results of the procedure. We notice that generally we do not benefit from discrete versions of hyperbolic tessellations, that is why we decided to exclude them from the further analysis. In the case of trees, we notice that the discrete version yields significantly better results for greedy success rate – for that reason we keep that tessellation. Finally, we excluded angular version of product geometry $\mathbb{H}^2 \times \mathbb{R}$ – we did not notice systematic gains in comparison to the non-angular version.

## B   ROBUSTNESS

In this appendix, we will explain how the issues mentioned in Section 8 were combated. We will also check if they could affect our results. Additionally, we have studied alternative methods of measuring distances, based on discrete tessellation distances and angular sizes .

**Possibly insufficient size of grids.**   For the sake of comparability, we aimed at keeping the number of neurons as close to 20,000 as possible. However, one could argue if this is enough. To combat the first two issues, in some geometries we consider coarser and finer grids: coarser grids are better at handling the first issue, and finer grids are better at handling the second issue – in both cases, we expect that increasing $d_R$ and grid density beyond some threshold yields diminishing returns. That is why, based on the results from the previous sections, we have added the so-called *big* versions – coarser but larger grids ($M = 100000$) – for selected, primising manifolds ($\mathbb{H}^3$, $\mathbb{H}^3*$, $\mathbb{H}^2 \times \mathbb{R}$, Solv, and Twist). We will denote them with **. See Table 8 for the details.

| name | dim | geometry | closed | nodes | diameter | description of the set $D$ |
|---|---|---|---|---|---|---|
| $\mathbb{H}^3**$ | 3 | hyperbolic | F | 100427 | 233 | $\{4, 3, 5\}$ hyperbolic honeycomb |
| $(\mathbb{H}^3*)**$ | 3 | hyperbolic | F | 100641 | 179 | $\{4, 3, 5\}$ subdivided(2) |
| Twist** | 3 | twist | F | 101230 | 184 | twisted $\{5, 4\} \times \mathbb{Z}$ |
| $\mathbb{H}^2 \times \mathbb{R}**$ | 3 | product | F | 100030 | 282 | bitruncated $\{7, 3\} \times \mathbb{Z}$ |
| Solv | 3 | solv | F | 100041 | 310 | as in Kopczyński & Celińska-Kopczyńska (2020) |

Table 8: Details on tessellations used in our study (*big* versions); * denotes finer grids.

We started by checking if there are significant differences in favour of *big* versions of manifolds; to this end we performed Wilcoxon tests with Bonferroni correction for multiple testing. Figure 3 depicts the results of the procedure. According to our results, in most of the cases the differences are insignificant, which suggests that the size of the manifold is not a severe threat to validity. Usage of *big* versions usually results in better embeddings for Rat connectomes; that might be correlated with a different function of those connectomes in comparison to others in the sample (they describe nervous systems). Rarely, *big* versions yield worse embeddings than the standard ones – usually for Human connectomes; however, no pattern enabling explanation is noticable here.

Next, we checked if the size of the manifolds affects rankings. To this end, we computed weighted Cohen's kappas (Cohen, 1968). In kappas, 0 represents the amount of agreement that can be expected from random chance, and 1 signifies perfect agreement between the raters. Originally, kappas take into account only agreements of the raters. The the weighted kappas allow disagreements to be weighted differently which is more suitable for us – we are more interested in the relative placement of the pairs of the geometries in the ranking than in the actual places. E.g., if there are small differences in ranks by two raters, e.g., by one, the ranks should remain similar to us as embeddings yielding results of the comparable quality should be still close to each other. Although there are no universal guidelines for the interpretation of those coefficients, the literature suggests that the values over 0.61 suggest moderate to substantial agreement between raters and values exceeding 0.81 – strong to almost perfect agreement (Landis & Koch, 1977).

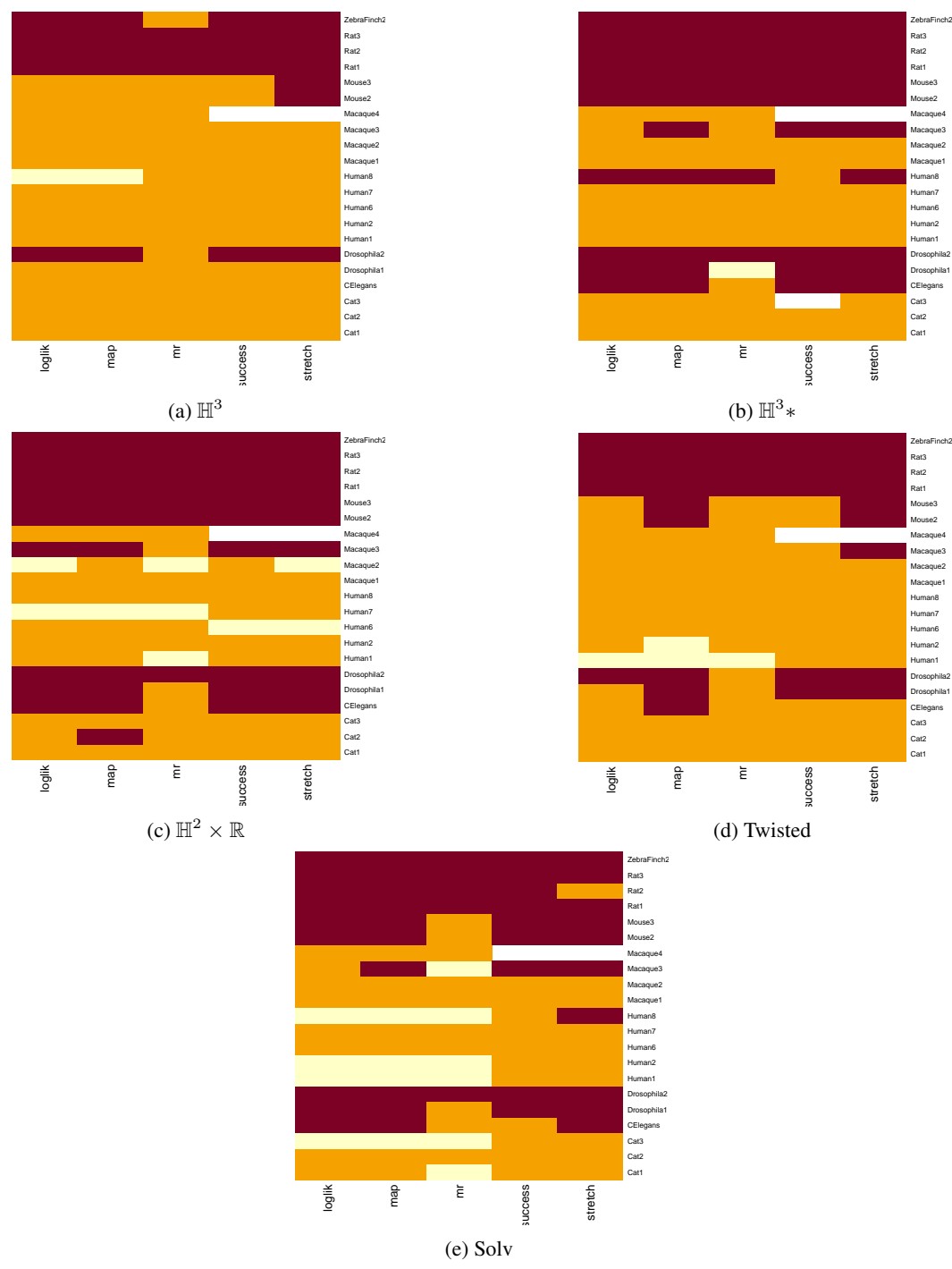

Figure 3: Comparison of the goodness-of-fit between regular and *big* versions of manifolds. Red suggests that the *big* version yields better results and the difference is significant; orange suggests lack of significant difference, and yellow suggests significantly worse results for *big* version.

Accroding to data in Table 9 rankings obtained from *big* versions of manifolds in the standard setup of Simulated Annealing ($N_s = 10,000$ iterations) are at least in substantial agreement with rankings based on standard versions. The high agreement in rankings based on voting rules is unsurprising. It is in line with the results depicted in Figure 3 – we include more information from the distributions,

| Pair of rankings | Max performance | | | | | Copeland | | | | |
|---|---|---|---|---|---|---|---|---|---|---|
| | NLL | MAP | IMR | SC | IST | NLL | MAP | IMR | SC | IST |
| Standard$_{SA:10,000}$ vs Big$_{SA:10,000}$ | 0.80 (0.72;0.88) | 0.75 (0.66;0.84) | 0.84 (0.77;0.90) | 0.57 (0.40;0.73) | 0.61 (0.47;0.74) | 0.86 (0.79;0.93) | 0.81 (0.74;0.88) | 0.91 (0.87;0.95) | 0.71 (0.61;0.82) | 0.78 (0.69;0.88) |
| Standard$_{SA:100,000}$ vs Big$_{SA:100,000}$ | 0.78 (0.69;0.88) | 0.75 (0.65;0.85) | 0.85 (0.79;0.91) | 0.52 (0.34;0.70) | 0.65 (0.53;0.77) | 0.85 (0.77;0.92) | 0.83 (0.76;0.89) | 0.84 (0.77;0.91) | 0.75 (0.66;0.84) | 0.82 (0.73;0.90) |
| Standard$_{SA:10,000}$ vs Standard$_{SA:100,000}$ | 0.95 (0.92;0.97) | 0.93 (0.90;0.96) | 0.93 (0.90;0.96) | 0.86 (0.80;0.92) | 0.87 (0.81;0.92) | 0.84 (0.77;0.90) | 0.82 (0.74;0.89) | 0.83 (0.76;0.91) | 0.77 (0.66;0.88) | 0.83 (0.75;0.91) |
| Big$_{SA:10,000}$ vs Big$_{SA:100,000}$ | 0.76 (0.66;0.86) | 0.70 (0.59;0.82) | 0.78 (0.69;0.87) | 0.73 (0.59;0.86) | 0.60 (0.44;0.75) | 0.79 (0.70;0.88) | 0.75 (0.66;0.84) | 0.86 (0.80;0.91) | 0.69 (0.55;0.83) | 0.76 (0.66;0.87) |
| Standard$_{SA:10,000}$ vs Big$_{SA:100,000}$ | 0.74 (0.63;0.85) | 0.74 (0.64;0.83) | 0.79 (0.70;0.89) | 0.51 (0.35;0.68) | 0.68 (0.56;0.8) | 0.76 (0.67;0.85) | 0.76 (0.67;0.85) | 0.82 (0.75;0.90) | 0.58 (0.44;0.72) | 0.79 (0.69;0.89) |

Table 9: The agreements between the rankings obtained for different simulation setups (values of Cohen's kappa). Standard includes: $\mathbb{H}^3$, $\mathbb{H}^3*$, $\mathbb{H}^2 \times \mathbb{R}$, Solv, and Twist. 95% confidence intervals in brackets.

so the results should be more robust than those based on max performance (outliers). However, we recommend treating the results with caution for greedy routing success and stretch.

**Possibly insufficient number of iterations for Simulated Annealing.** As Simulated Annealing is a probabilistic technique for approximating the global optimum of a given function, one could argue that our results could be improved by increasing, e.g., the number of iterations (the third issue). While in the main paper, we describe the results obtained with Simulated Annealing with $N_s = 10,000$ iterations per simulation iteration, we also checked if our results differ if we perform Simulated Annealing with $N_s = 100,000$ iterations per simulation iteration instead. As expected, for loglikelihood, MAP, and MR we cannot reject the hypotheses that the results obtained with larger number of iterations are usually better. However, surprisingly, for greedy success rate and stretch the results worsen with the increase in the number of iterations (Figure **??** depicts the results of Wilcoxon tests with Bonferroni correction for multiple testing).

We checked if the number of iterations for Simulated Annealing $N_s$ affected our results regarding rankings with Cohen's kappas. Based on the data in Table 9, we notice that pairs of rankings are in at least substantial agreement. Especially, the results regarding standard grids (presented in the main part of the paper) are robust to $N_s$ – the values of kappas for optimistic scenario are over 0.85 and for rankings based on voting rules they usually exceed 0.80. Although the agreements of rankings if we just change $N_s$ for *big* grids are still satisfying (most values of kappas over 0.75), the results from comparison of rankings based on standard grids with shorter time for Simulated Annealing against the *big* grids with longer time for Simulated Annealing suggests that the *big* versions of grids might be affected by $N_s$. Again, we notice that greedy routing success rate and stretch seem to be less immune to the setup of Simulating Annealing, that is why we suggest caution while generalizing the results obtained for them.

The size of the grid or the setup of the Simulated Annealing may affect the results. In some experiments, standard versions of the grids have obtained significantly better results than the so-called *big* versions. Since the difference between these two cases is that the *big*-variant has simply larger number of cells, this should not happen, since any embedding in the standard variant is also an embedding in the *big*-variant. This seems to be caused either by a failure to correctly guess the optimal values of the parameters, or possibly by Simulated Annealing requiring more iterations to find good embeddings on larger distances.

**Alternative methods of obtaining $R$ and $T$** The fourth issue is challenging. As explained in Section 5, the values of $R$ and $T$ have been obtained by dynamically adjusting them during the simulated annealing process (A). We have also experimented with other methods: $R$ is adjusted but $T$ remains fixed (B), and both $R$ and $T$ remains fixed. We run 30 iterations using method (A), then 30 iterations using method (B), then 30 iterations using method (C). The fixed values of $R$ and $T$ are based on the best result (by loglikelihood) obtained in the earlier iterations.

If the methods change the results, we should notice level shifts in the time series of the values of the quality measures – level shift appears as a parallel movement of the trend line. That is why we started with identification of possible locations of the level shifts in our results. Most of the time series (identified by a pair animal and geometry) has two level shifts – around 30th and 60th iteration that correspond to the starting points of new methods (Figure 5).

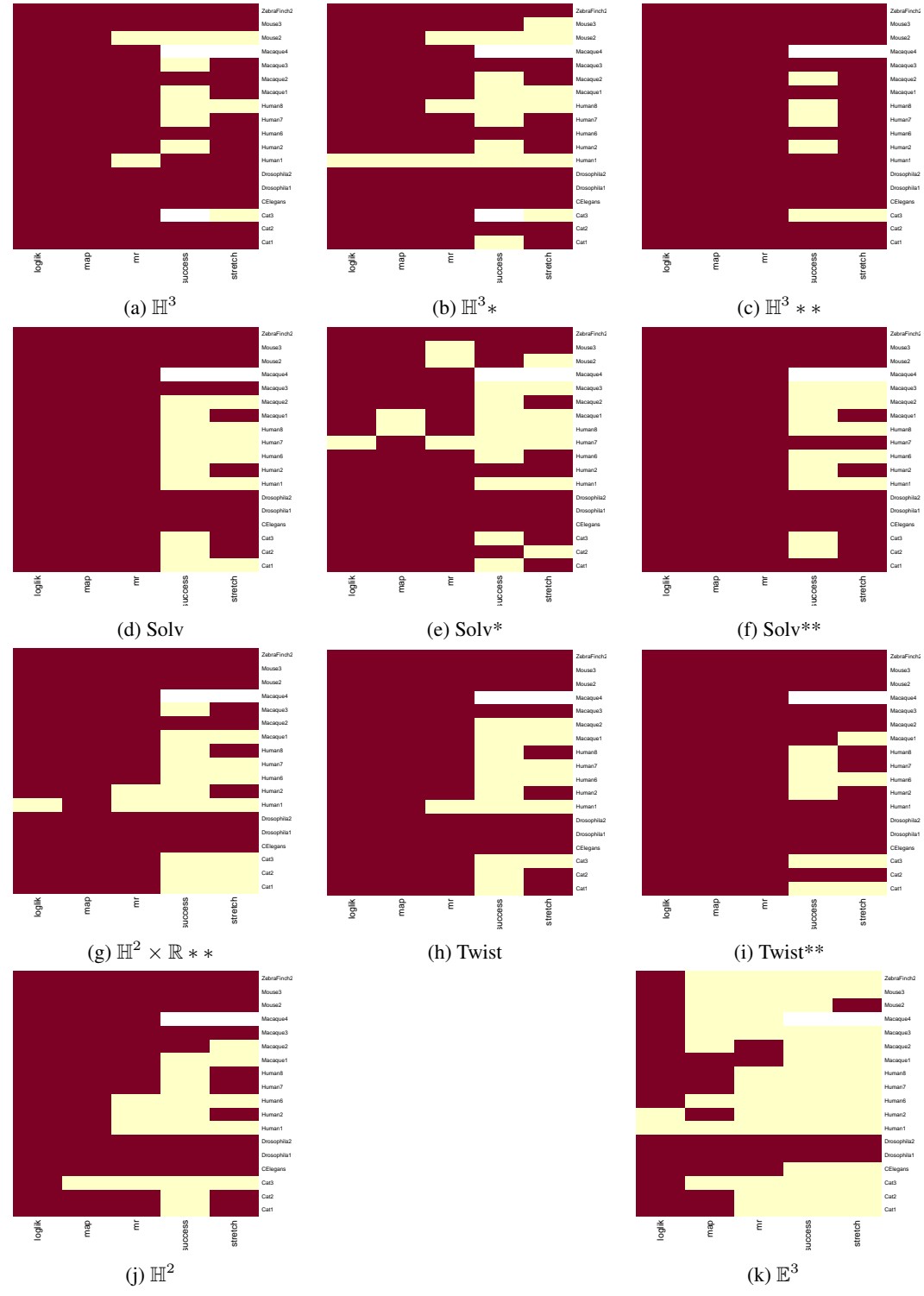

Figure 4: Comparison of the goodness-of-fit between results of Simulated Annealing with 10.000 vs. 100.000 steps per iteration. Red suggests that the longer version yields better results and the difference is significant; orange suggests lack of significant difference, and yellow suggests significantly worse results for longer version.

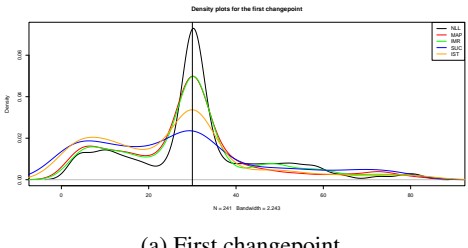
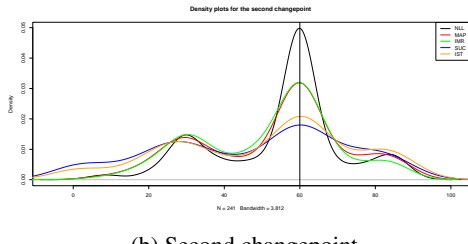

| (a) First changepoint | (b) Second changepoint |

Figure 5: Density plots for changepoints in time series of measures (indicators for level shifts)

To understand the impact of the change in method on the values of the quality measures, we use OLS regressions. We control for the characteristics of connectomes: number of nodes in the connectome $n$, number of edges in the connectome $m$, its *density*, *assortativity* and *clustering* coefficients, and the *zone* of the connectome; we also take into account the number of availabe *cells* in the grid and its *geometry*.

| | 100·NLL | | 100·mAP | | 100·IMR | | 100·SC | | 100·ISTR | |
|---|---|---|---|---|---|---|---|---|---|---|
| | Estimate | $P(>|t|)$ | Estimate | $P(>|t|)$ | Estimate | $P(>|t|)$ | Estimate | $P(>|t|)$ | Estimate | $P(>|t|)$ |
| (Intercept) | 4.737e+00 | 0.00 | 2.719e+01 | 0.00 | -4.323e+01 | 0.00 | 8.118e+01 | 0.00 | 6.930e+01 | 2.205e-01 |
| n | 4.876e-03 | 0.00 | -1.050e-02 | 0.00 | -1.561e-02 | 0.00 | -1.857e-02 | 0.00 | -1.799e-02 | 1.011e-04 |
| m | -6.801e-05 | 1.65e-11 | -1.553e-04 | 0.00 | 7.865e-05 | 1.81e-10 | 1.032e-04 | 0.00 | 6.741e-05 | 5.579e-06 |
| density | -2.428e-01 | 0.00 | 9.127e-02 | 0.00 | -8.497e-01 | 0.00 | 1.545e-01 | 0.00 | 8.551e-02 | 5.370e-03 |
| assort | 1.302e+01 | 0.00 | 3.379e+01 | 0.00 | -1.687e+01 | 0.00 | 1.431e+01 | 0.00 | 3.645e+00 | 1.991e-01 |
| cluster | 8.659e+01 | 0.00 | 8.163e+01 | 0.00 | 1.428e+02 | 0.00 | 1.539e+01 | 0.00 | 2.998e+01 | 2.854e-01 |
| nervous | -1.061e+00 | 5.37e-07 | -6.122e+00 | 0.00 | -3.455e+01 | 0.00 | 1.285e+00 | 0.00 | -2.679e+00 | 1.169e-01 |
| other | 1.564e+00 | 0.00 | -3.702e+00 | 0.00 | -2.141e+00 | 0.00 | -3.542e+00 | 0.00 | -3.940e-01 | 9.186e-02 |
| cells | 2.347e+00 | 2.94e-15 | 4.730e+00 | 0.00 | 3.912e+01 | 0.00 | 1.425e+00 | 2.36e-15 | 4.654e-01 | 1.642e-01 |
| hyperbolic | 8.233e+00 | 0.00 | 9.441e+00 | 0.00 | -3.735e+01 | 0.109924 | 8.125e+00 | 0.00 | 7.031e+00 | 1.057e-01 |
| other | 9.250e+00 | 0.00 | 9.527e+00 | 0.00 | 3.351e+00 | 0.202089 | 6.862e+00 | 0.00 | 6.473e+00 | 1.189e-01 |
| product | 8.502e+00 | 0.00 | 7.357e+00 | 0.00 | 7.195e+00 | 0.006128 | 6.931e+00 | 0.00 | 6.187e+00 | 1.188e-01 |
| solv | 7.399e+00 | 0.00 | 6.787e+00 | 0.00 | 8.688e-01 | 0.000372 | 5.140e+00 | 0.00 | 4.747e+00 | 1.104e-01 |
| nodes | 3.950e-05 | 0.00 | 4.645e-05 | 0.00 | 1.631e-05 | 0.00 | 2.202e-05 | 0.00 | 2.445e-05 | 6.950e-07 |
| B | -7.985e-01 | 5.76e-12 | -5.009e-01 | 4.18e-07 | -2.787e-01 | 0.049061 | -4.759e-01 | 1.19e-11 | -5.433e-01 | 6.406e-02 |
| C | 5.925e-01 | 3.30e-07 | 4.841e-01 | 1.02e-06 | 6.684e-01 | 2.41e-06 | 3.999e-01 | 1.23e-08 | 3.746e-01 | 6.412e-02 |
| $R^2$ (adjusted $R^2$) | 0.8061 (0.806) | | 0.8948 (0.8948) | | 0.8426 (0.8245) | | 0.8194 (0.8193) | | 0.8784 (0.8784) | |
| p-value for F-test | 0.00 | | 0.00 | | 0.00 | | 0.00 | | 0.00 | |

Table 10: OLS regression results for the determinants of the quality measures. Number of observations = 24,536.

For all the quality measures, we notice that on average, the method B leads to lower values of the respective quality measures, and the method C increases the values of the respective quality measures in comparison to results obtained with method A, ceteris paribus (Table **??**). The differences are statistically significant. However, even if we are aware that with the increase in the number of observations, the p-values drop to zero, we work here with the multilevel categorical variables, so we are unable to comment on the size of effect (available methods based on partial regressions and $R^2$ coefficients would allocate the impact to the constant term). The regressions have substantial explanatory power ($R^2$ coefficients at least 80%).

To sum up, if one is interested in optimizing the quality measures, we would recommend usage of the method B. We are fully aware that our choice of method A for Simulated Annealing may affect the final results, in particular, even our "best" evaluations may not be optimal. However, there are at least two advantages of our approach. First, allowing the algorithm to optimize does not favor any of the tessellations – all of them have the same chances to find an optimal solution. This way we ensure comparability of our results. We have observed that the first iteration may yield worse results due to the poor initially guessed values of $R$ and $T$; however, in the further experiments, while the initial values of $R$ and $T$ change, it does not affect the end results much. Eventually, we get data that is independently distributed, i.e., there is no serial correlation (all p-values in Ljung-Box test smaller than $10^{-5}$) which makes statistical analysis of the results significantly easier.

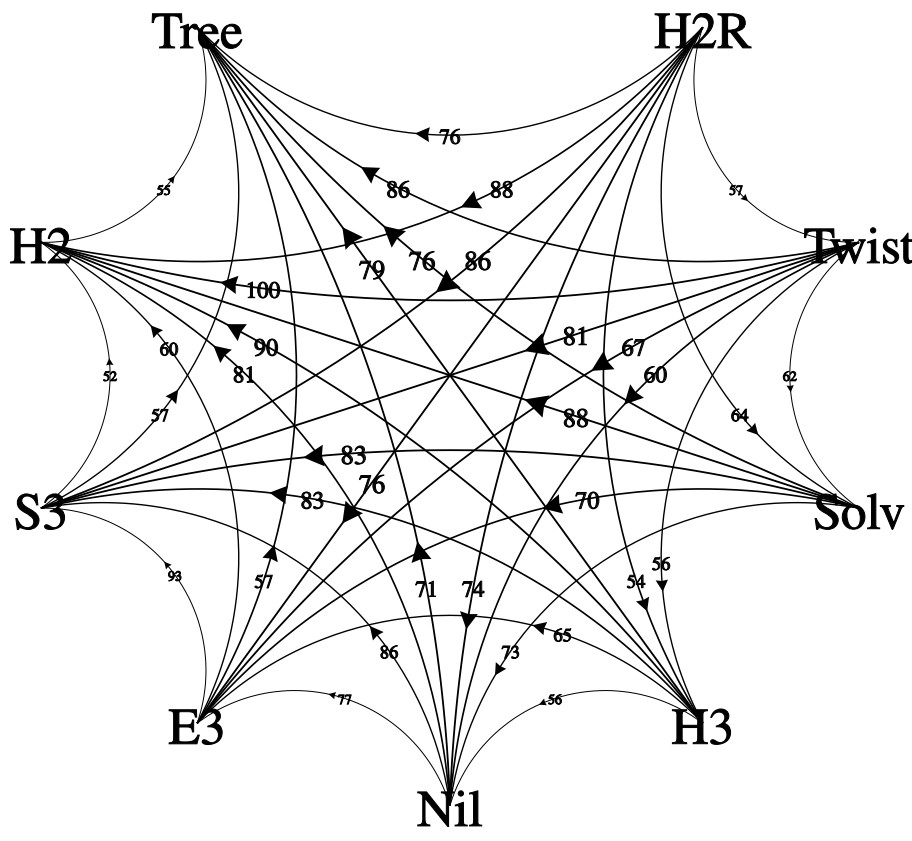

Figure 6: Normalized loglikelihood

## C  WEIGHTED NETWORKS

To allow for generalizations, in Figures 6-10 we provide weighted directed networks constructed upon the voting rules. The weights correspond to the percent of connectomes for which the source geometry in the edge beats the target geometry. Embeddings to Twist have 100% success rate over embeddings in $\mathbb{H}^2$ (for quality measures different than greedy routing success).

## D  DETAILED RANKINGS

Figures 11, 12, 15, 14, and 13 visualize the rankings of the tessellations. Table 11 show the descriptive statistics for ranks obtained by geometries.

## E  COMPARISON

Table 12 is the full version of Table 6. We list our results for four measures (we do not include loglikelihood since not all of these embedders are based on the maximum loglikelihood method).

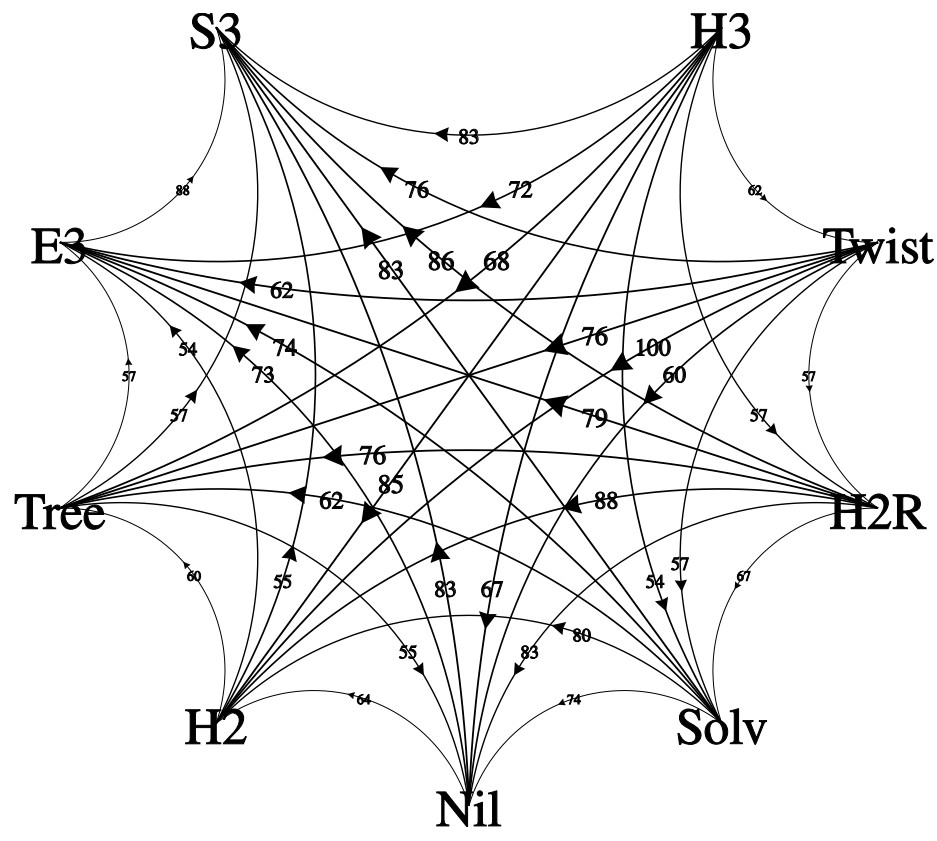

Figure 7: MAP

| geometry | MIN | | | | | MED | | | | | MAX | | | | |
|---|---|---|---|---|---|---|---|---|---|---|---|---|---|---|---|
| | NLL | MAP | IMR | SC | IST | NLL | MAP | IMR | SC | IST | NLL | MAP | IMR | SC | IST |
| $\mathbb{H}^2$ | 2 | 2 | 2 | 2 | 8 | 4 | 10 | 2 | 13 | 9 | 12 | 12 | 12 | 15 | 13 |
| $\mathbb{H}^2\&$ | 1 | 1 | 1 | 1 | 1 | 3 | 4 | 3 | 1 | 2 | 6 | 7 | 6 | 8 | 8 |
| tree | 1 | 1 | 1 | 1 | 8 | 3 | 9 | 1 | 13 | 10 | 15 | 15 | 14 | 15 | 15 |
| $\mathbb{E}^3$ | 2 | 2 | 2 | 1 | 3 | 5 | 3 | 6 | 5 | 5 | 14 | 15 | 15 | 13 | 15 |
| $\mathbb{E}^3\&$ | 2 | 2 | 2 | 1 | 1 | 5 | 3 | 10 | 3 | 3 | 14.5 | 14.5 | 15 | 9 | 14 |
| $\mathbb{H}^3$ | 4 | 4 | 3 | 8 | 6 | 12 | 13 | 6 | 11 | 12 | 15 | 15 | 15 | 15 | 15 |
| $\mathbb{H}^3*$ | 9 | 8 | 5 | 6 | 7 | 12 | 13 | 10 | 12 | 14 | 15 | 15 | 15 | 15 | 15 |
| $\mathbb{H}^3\&$ | 5 | 5 | 5 | 2 | 2 | 7 | 6 | 8 | 2 | 5 | 14 | 15 | 15 | 8 | 15 |
| Nil | 4 | 4 | 5 | 5 | 5 | 8 | 7 | 9 | 7 | 8 | 14 | 13 | 15 | 11 | 10 |
| Nil* | 4 | 4 | 4 | 4 | 5 | 7 | 5 | 11 | 6 | 6 | 14.5 | 14.5 | 15 | 10.5 | 14 |
| Twist | 4 | 4 | 4 | 5 | 4 | 13 | 13 | 10 | 11 | 13 | 15 | 14 | 14 | 14 | 15 |
| $\mathbb{H}^2 \times \mathbb{R}$ | 8 | 8 | 7 | 8 | 8 | 12 | 11 | 12 | 10 | 11 | 15 | 15 | 15 | 12 | 15 |
| Solv | 5 | 4 | 4 | 4 | 5 | 11 | 10 | 8 | 10.5 | 10 | 15 | 15 | 15 | 14 | 15 |
| Solv* | 7 | 7 | 8 | 6 | 7 | 10 | 8 | 11 | 8 | 8 | 15 | 15 | 14 | 11 | 13 |
| $\mathbb{S}^3$ | 1 | 1 | 1 | 1 | 1 | 2 | 1 | 4 | 4 | 2 | 9 | 15 | 15 | 9 | 9 |

Table 11: Descriptive statistics (minimum, median, maximum) for ranks obtained by geometries (at the maximum performance)

| connectome | dim | mAP | method | rad | time | ours | better |
|---|---|---|---|---|---|---|---|
| celegans | 2 | 0.500 | Poincaré | 7.2 | 278 | 0.540 | 30 |
| celegans | 3 | 0.583 | Poincaré | 10.1 | 274 | 0.584 | 21 |
| drosophila1 | 2 | 0.425 | Mercator (full) | 23.6 | 14 | 0.483 | 30 |
| drosophila1 | 3 | 0.488 | Poincaré | 11.4 | 365 | 0.512 | 30 |
| human1 | 2 | 0.651 | Lorentz | 10.8 | 1085 | 0.675 | 30 |
| human1 | 3 | 0.722 | Poincaré | 9.4 | 827 | 0.799 | 30 |
| mouse3 | 2 | 0.585 | Mercator (full) | 29.9 | 117 | 0.612 | 30 |
| mouse3 | 3 | 0.654 | Poincaré | 12.2 | 9207 | 0.655 | 18 |
| connectome | dim | MeanRank | method | rad | time | ours | better |
| celegans | 2 | 39.5 | BFKL | 7.8 | 0 | 30.1 | 30 |
| celegans | 3 | 27.3 | Poincaré | 9.9 | 277 | 26.3 | 29 |
| drosophila1 | 2 | 54.4 | BFKL | 8.2 | 1 | 45.0 | 30 |
| drosophila1 | 3 | 39.9 | Poincaré | 11.6 | 354 | 37.1 | 29 |
| human1 | 2 | 43.1 | Poincaré | 11.9 | 1284 | 38.6 | 23 |
| human1 | 3 | 26.9 | Poincaré | 9.5 | 835 | 17.8 | 30 |
| mouse3 | 2 | 103.5 | Mercator (fast) | 29.1 | 87 | 92.4 | 30 |
| mouse3 | 3 | 84.6 | Poincaré | 12.2 | 9207 | 78.5 | 29 |
| connectome | dim | success | method | rad | time | ours | better |
| celegans | 2 | 0.903 | Poincaré | 7.2 | 267 | 0.931 | 27 |
| celegans | 3 | 0.958 | Poincaré | 10.1 | 274 | 0.930 | 0 |
| drosophila1 | 2 | 0.769 | Mercator (full) | 23.6 | 14 | 0.847 | 30 |
| drosophila1 | 3 | 0.844 | Poincaré | 11.4 | 365 | 0.843 | 13 |
| human1 | 2 | 0.889 | Poincaré | 12.2 | 1185 | 0.929 | 21 |
| human1 | 3 | 0.926 | Poincaré | 9.5 | 835 | 0.958 | 24 |
| mouse3 | 2 | 0.962 | Mercator (full) | 34.5 | 74 | 0.967 | 30 |
| mouse3 | 3 | 0.971 | Poincaré | 12.2 | 8679 | 0.952 | 0 |
| connectome | dim | stretch | method | rad | time | ours | better |
| celegans | 2 | 1.970 | Mercator (fast) | 16.4 | 6 | 1.254 | 30 |
| celegans | 3 | 1.230 | Poincaré | 9.5 | 268 | 1.232 | 16 |
| drosophila1 | 2 | 2.270 | Mercator (fast) | 23.1 | 13 | 1.340 | 30 |
| drosophila1 | 3 | 1.360 | Poincaré | 10.3 | 354 | 1.328 | 30 |
| human1 | 2 | 2.030 | Mercator (fast) | 20.7 | 2 | 1.282 | 30 |
| human1 | 3 | 1.240 | Poincaré | 9.3 | 812 | 1.176 | 28 |
| mouse3 | 2 | 1.260 | Mercator (fast) | 28.9 | 40 | 1.156 | 30 |
| mouse3 | 3 | 1.080 | Poincaré | 12.2 | 9207 | 1.145 | 0 |

Table 12: Our embeddings versus state-of-the-art. For each connectome and dimension, we list the best prior method and its result, the radius of the embedding, time elapsed in seconds, the best result of our method, and how many times (out of 30) our result was better.

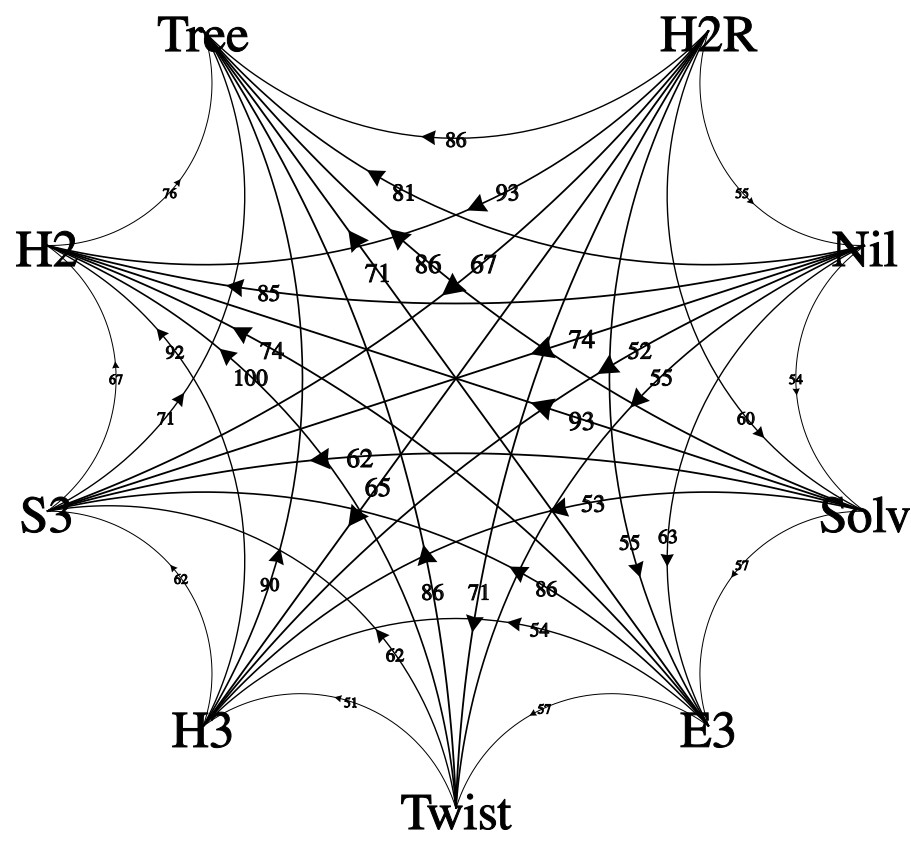

Figure 8: IMR

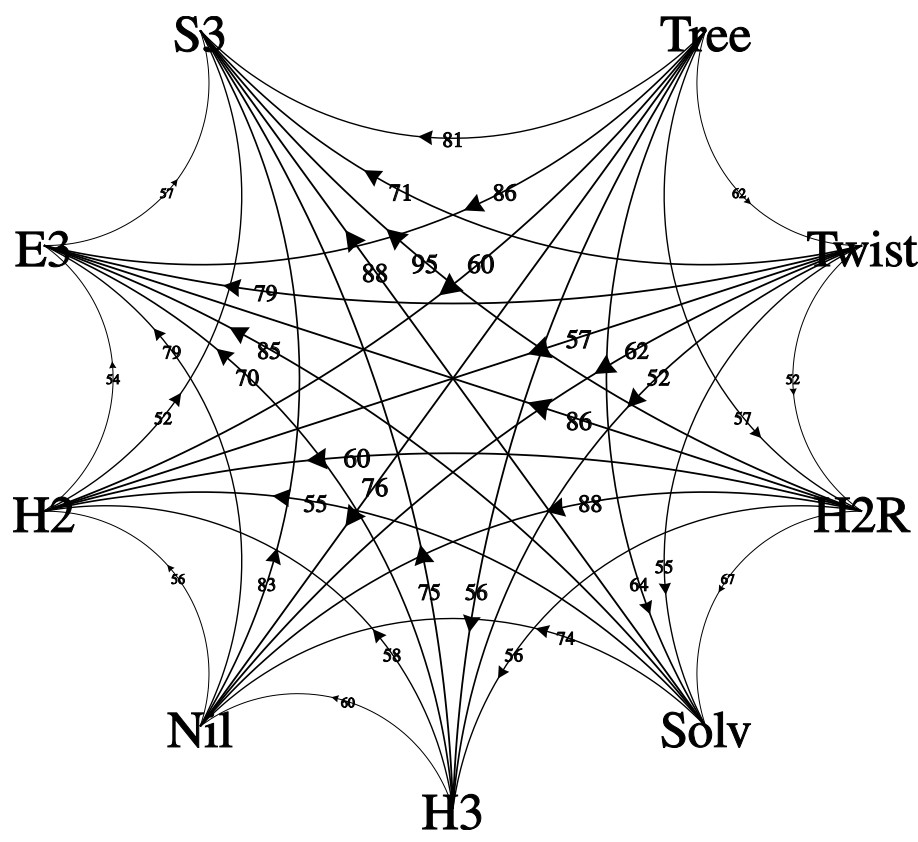

Figure 9: SC

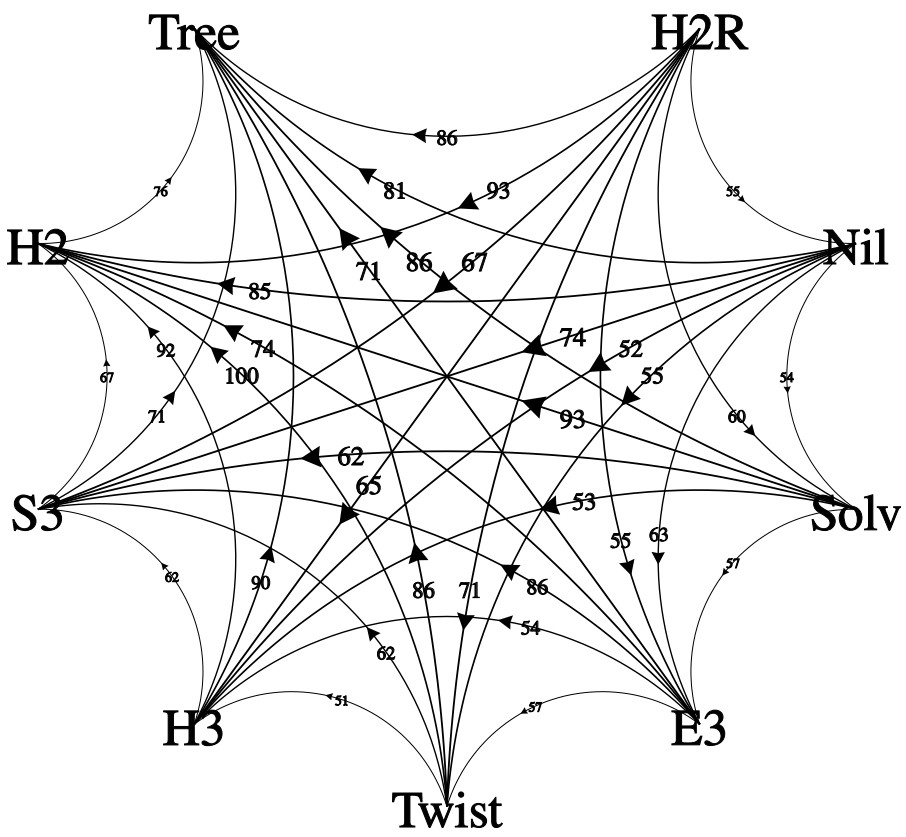

Figure 10: ISTR

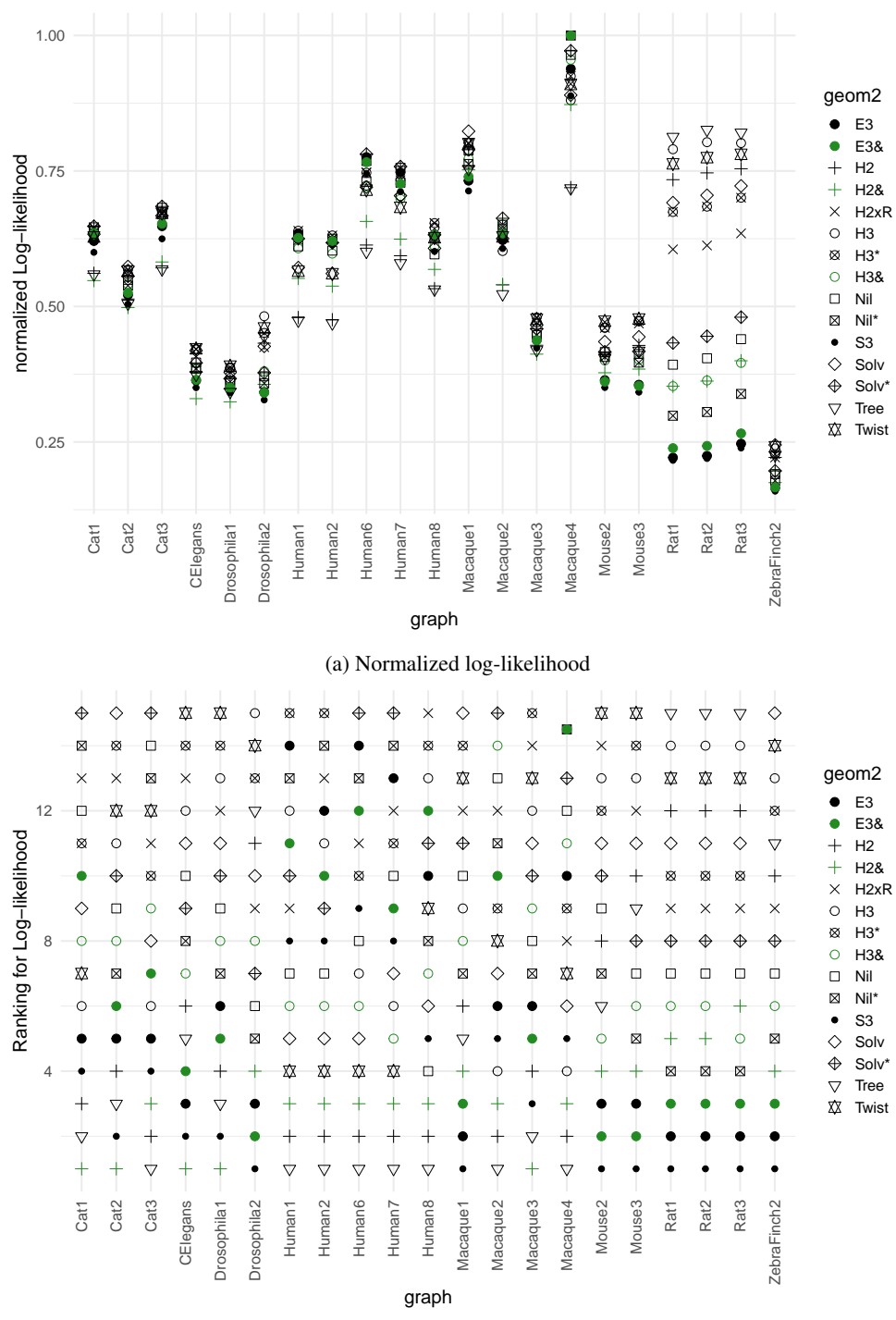

(a) Normalized log-likelihood

(b) Normalized log-likelihood – ranks

Figure 11: Our best embeddings – log-likelihood. Top = best embedding obtained, bottom = worst embedding obtained, * = fine grid.

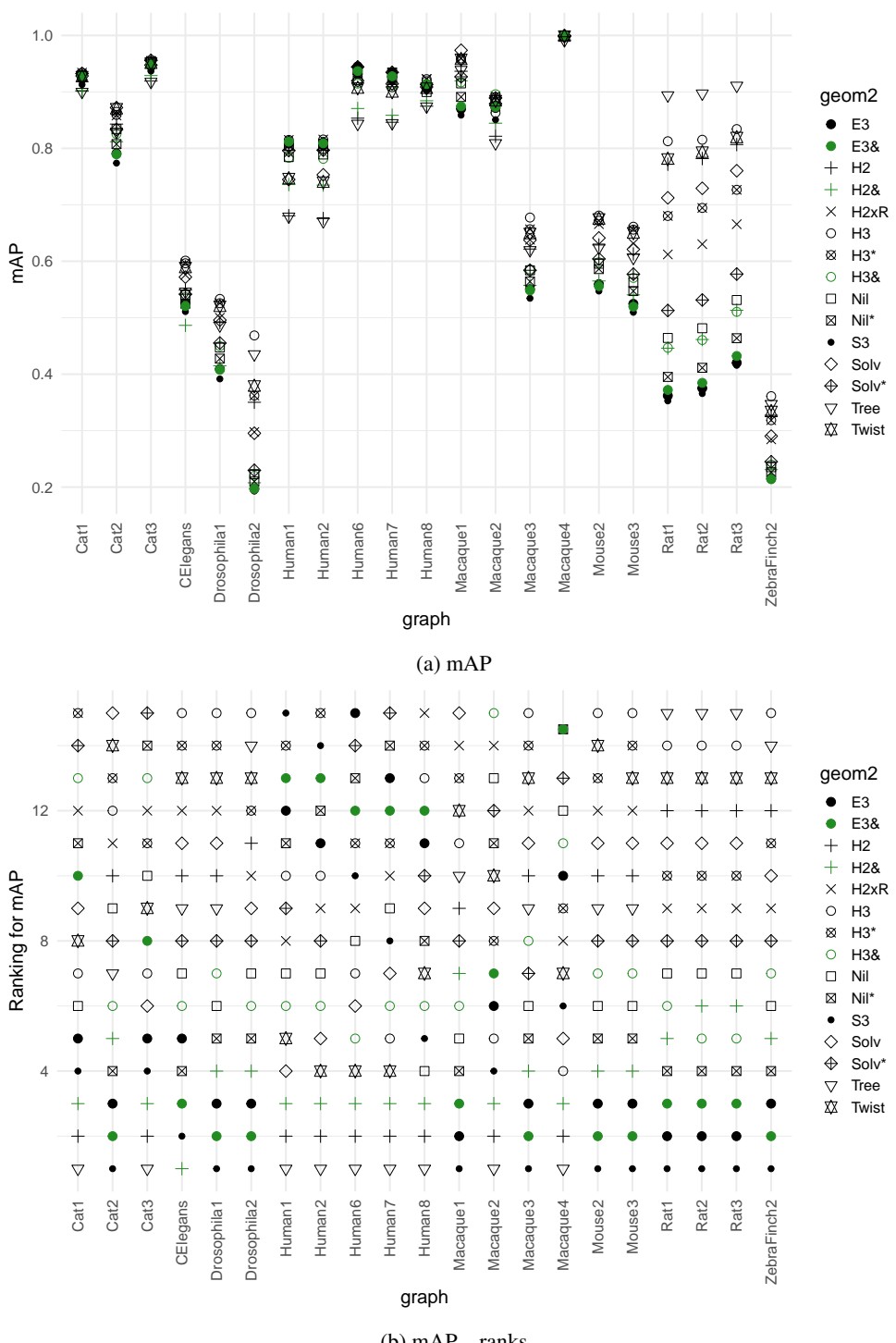

(a) mAP

(b) mAP – ranks

Figure 12: Our best embeddings – mAP. Top = best embedding obtained, bottom = worst embedding obtained, * = fine grid.

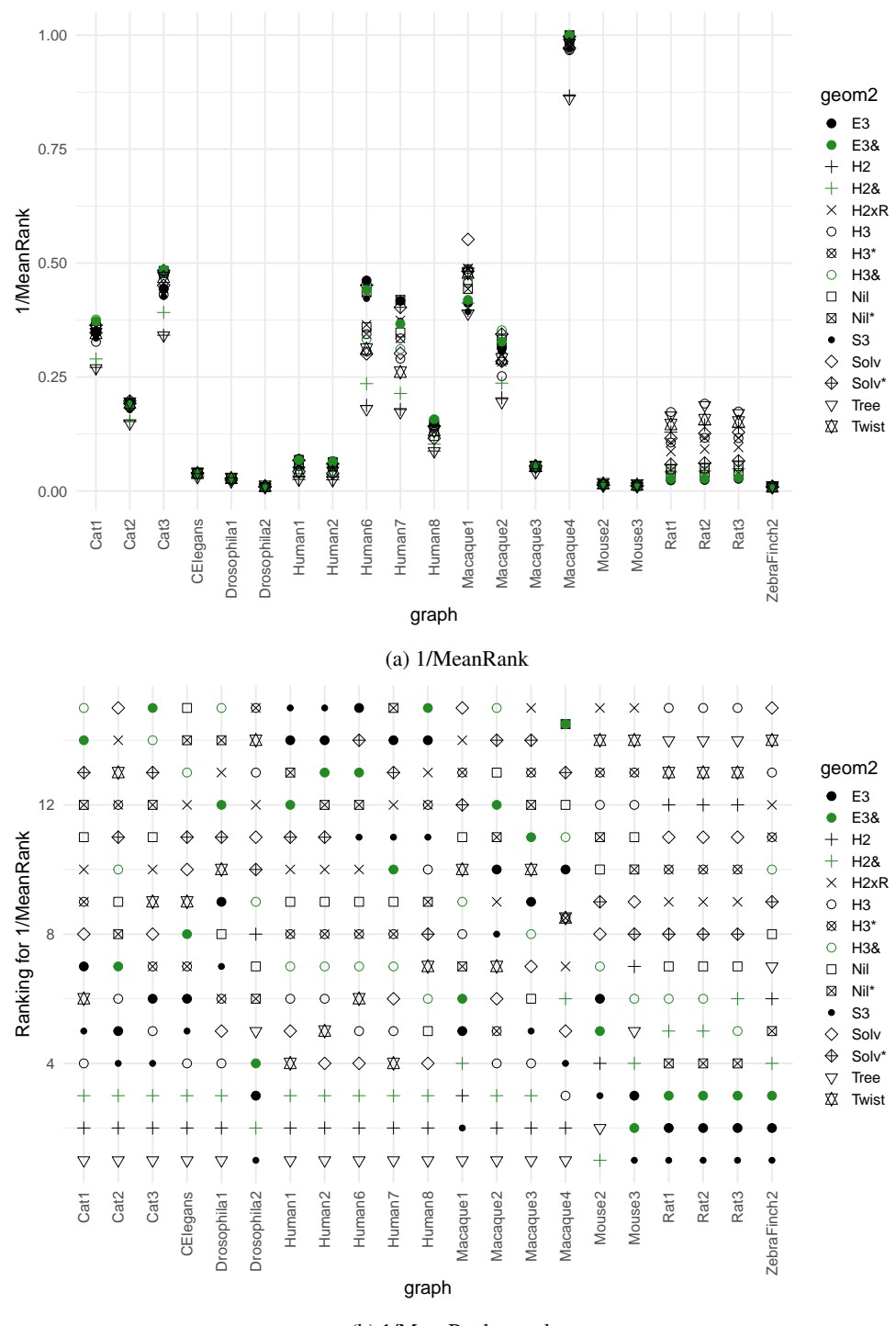

(a) 1/MeanRank

(b) 1/MeanRank – ranks

Figure 13: Our best embeddings – MeanRank. Top = best embedding obtained, bottom = worst embedding obtained, * = fine grid.

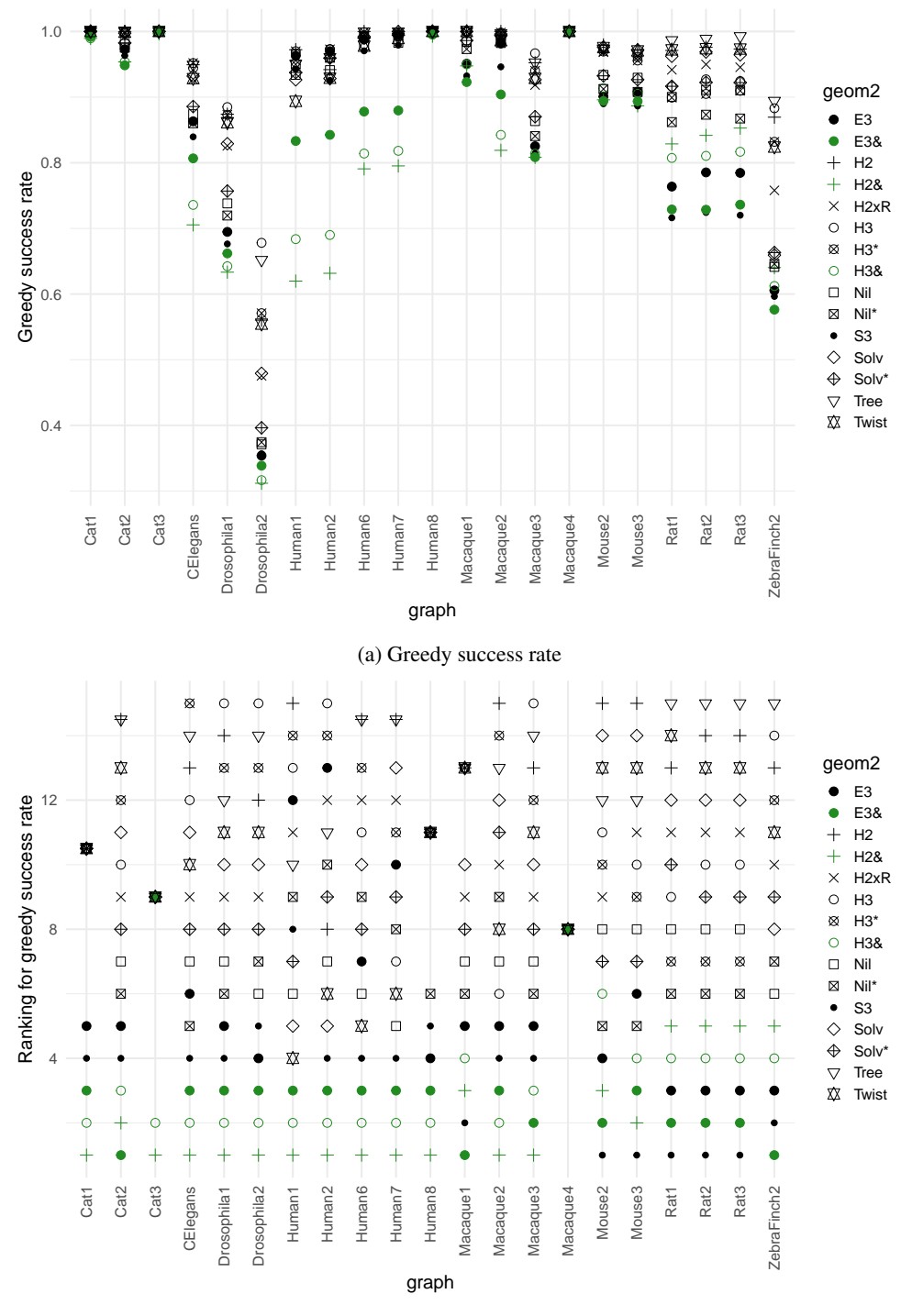

(a) Greedy success rate

(b) Greedy success rate – ranks

Figure 14: Our best embeddings – greedy success rate. Top = best embedding obtained, bottom = worst embedding obtained, * = fine grid.

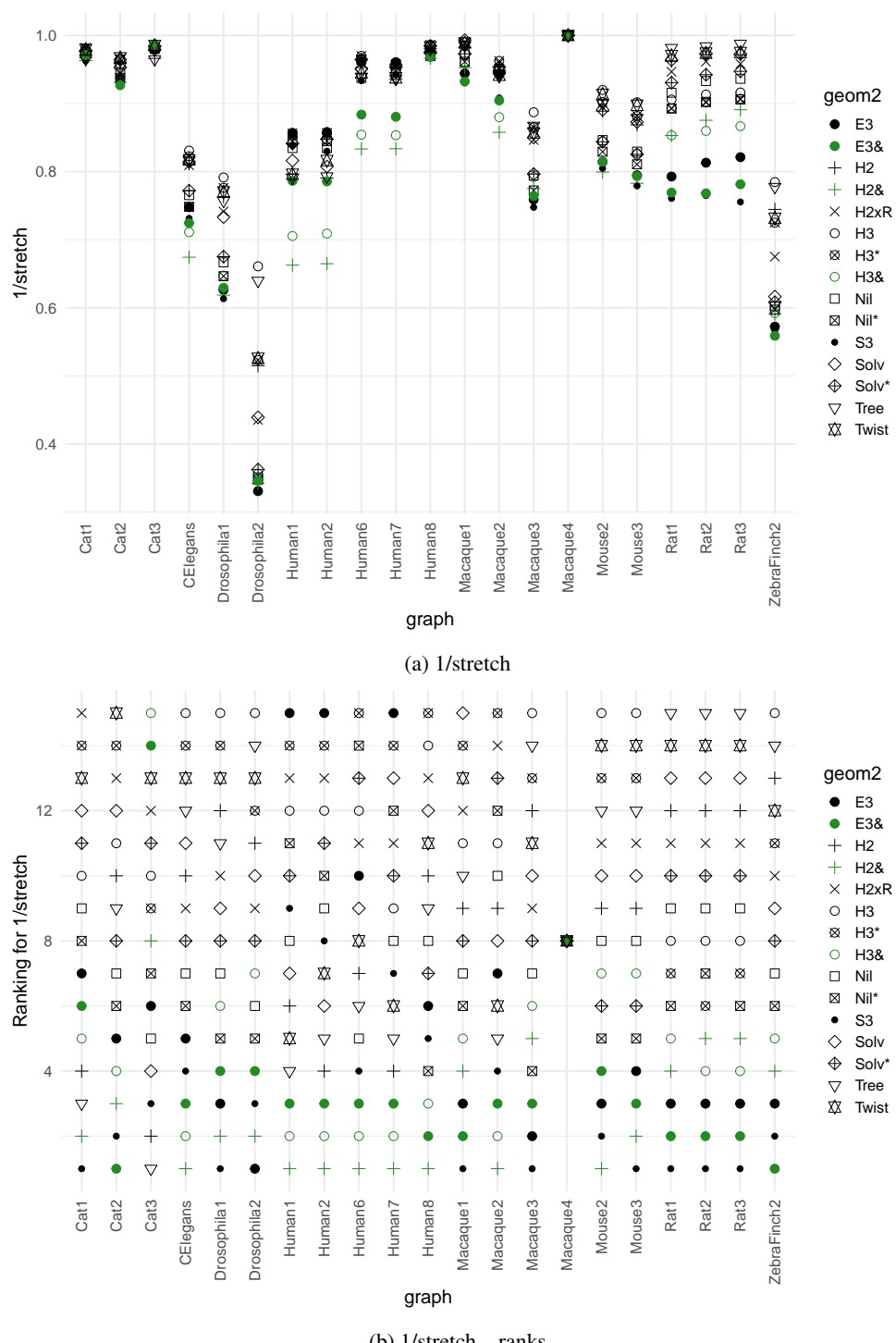

(a) 1/stretch

(b) 1/stretch – ranks

Figure 15: Our best embeddings – stretch. Top = best embedding obtained, bottom = worst embedding obtained, * = fine grid.

