# OpenReview forum: "Modelling brain connectomes networks: Solv is a worthy competitor to hyperbolic geometry!"
_ICLR.cc/2024/Conference — Submitted to ICLR 2024_

### Official Review · Reviewer_7zXB · 2023-11-01

**Soundness:** 3 good
**Presentation:** 2 fair
**Contribution:** 3 good
**Rating:** 6
**Confidence:** 2

**Summary:**

This paper studies an interesting problem, that is, embedding the brain connectome to some kind of geometry. To this end, the paper proposes an embedding algorithm based on Simulating Annealing that allows the embedding of connectomes to Euclidean, Spherical, Hyperbolic, Solv, Nil, and also product geometries.

**Strengths:**

An interesting topic about modeling brain connectome networks, which can facilitate the study of brain functions and learning mechanisms.

**Weaknesses:**

The paper may miss some discussion with Euclidean geometry, which the ML community may be more familiar with.
For example, in Euclidean geometry, Voronoi tessellation has recently been successfully used to study the mouse brain connectome (in "Network structure of the mouse brain connectome with voxel resolution, Science Advances 2020"). Moreover, there are even more kinds of tessellations beyond Voronoi in Euclidean geometry. Have the authors considered this possibility?

On the other hand, network/graph/node embedding has been extensively studied by the deep learning community. I wonder is it possible to apply some graph neural network methods, such as node2vec ("node2vec: Scalable Feature Learning for Networks, SIGKDD") for this problem?

The writing and presentation can still be largely improved. For instance, there are multiple typo or grammar issues and it can be better to make the paper more accessible by the machine learning community.

**Questions:**

Can the authors provide more comparisons (and possibly illustrations) to Euclidean embedding?

In some cases, the proposed method is never better than the previous state-of-the-art, e.g., on celegans with 3-dim (0 out of 30) and on mouse3 with 3-dim (0 out of 30). Is there any analysis of the failure cases?

---

### Official Review · Reviewer_Qkfi · 2023-11-04

**Soundness:** 2 fair
**Presentation:** 1 poor
**Contribution:** 2 fair
**Rating:** 3
**Confidence:** 3

**Summary:**

This paper presents a simulated annealing based embedding approach for modeling brain connectomes from across different species. This allows modeling of geometries arising from non-euclidean structure- specifically spherical, hyperbolic, solv, nil and other product geometries. The main claims of the paper is that this algorithm is (1) more suitable for finding embeddings in all of considered cases, (2) that resulting solv embeddings providing reasonable performance compared to alternatives and (3) the first of its kind to compare connectome embeddings across different Thurston geometries.

Experimental comparisons are performed on simulated and real-world connectomic data. The quality of the resulting embeddings is compared against baseline algorithms on measures such as mean Average Precision (mAP), MeanRank, greedy routing success and stretch

**Strengths:**

The central idea of generalizing connectomic representations to Thurston geometries is novel and very interesting, as is the proposed universal method of constructing such representations. The ideas presented here could be very useful to advancing applications in this domain in the future. Extensive experiments have been performed on simulated and real world connectomes from different species as a comparison, which is a plus.

**Weaknesses:**

1. The style of presentation of content in this manuscript makes it very challenging to follow for readers without the requisite background in topology.

(a) For example, section 3 Thurston Geometry introduced a lot of jargon and notation without definition -  "universal cover" / $\text{SL}(2,\mathcal{R}).

(b) Similarly, comparing  the caption description of Fig 1 with the textual description in paragraph 3 of section 3 makes it more difficult and abstract than necessary to understand what the authors are actually trying to convey (A simple fix would be to mark the corresponding points referred to in the text to illustrate the main point). Additionally, the tessalations in the figure are not actually referred to/described in the text using the terminology of the caption, which is very strange.

2. There is little to no background on connnectomics and/or representation learning for connectomics, beyond a few scattered citations. This is rather surprising, since this is a very active field with several works spanning diverse perspectives and approaches from graph theory, statistical models, deep learning, to name a few. The datasets/simulation parameters used in the paper are not described well enough to follow

3. The application aspect of the paper is ill motivated and kind of lost in the emphasis on mathematical explanation. The paper does not do a good job of illustrating why the embeddings are actually helpful for brain connectomics beyond Thurston's conjecture from 1982. This makes it really hard to appreciate why this approach is particularly useful for brain connectomes. In fact, it is very hard to discern what kind of connectomes- functional/structural the embeddings are being applied to.

4. It is not clear how the authors arrive at the likelihood expression in Section 4, which seemingly assumes independence in the pairwise interactions and a functional form for the probability based on the distance measure. Additionally, does the connectome model (V ,E) consider weighted or negative valued edges, as one would obtain from pairwise similarity measure in functional connectomics?

5. The datasets are not described at all in the main paper beyond the references in Table 1. The results in Tables 3 and 4 are really hard to parse and require the reader to go back and forth between the explanation and the table- since the captions are very nondescript, with best performances not highlighted. Additionally, no standard deviation measures have been reported (to quantify variability in either the simulation/and or population)

**Questions:**

It would be great if the authors could work on the following aspects of the paper:

(a) Motivating the contribution from the applications perspective
(b) Discussing the assumptions made by this embedding approach and why they are suitable/reasonable for this domain - for example, why three dimensional representations are sufficient for high dimensional data
(c) Contextualizing the work in light of other approaches used for representation learning
(d) Please have a table of notation for easy reference in the appendix and define any abbreviations/notation before usage
(e) Would be good to provide insight into the computation complexity of this approach -  how computationally expensive is the simulated annealing with Dijkstra's search in Section 4? how does this compare with other approaches? how long does the overall method take to converge?
(f) Please provide more details on how algorithmic parameters/ experiments are setup- eg. percentage of data used for computing the embeddings vs independent testing, number of iterations etc

---

### Official Review · Reviewer_wh8L · 2023-11-04

**Soundness:** 3 good
**Presentation:** 3 good
**Contribution:** 2 fair
**Rating:** 3
**Confidence:** 4

**Summary:**

This paper studies the visualization of connectomes which are comprehensive maps of neural connections in the brain. This conducts various experiments to compare the use of various geometries including hyperbolic,  Solv, Nil, and others on the embedding space.

**Strengths:**

- The study of the visualization of connectomes is important to understand cognitive processes.

- The paper conducts a comprehensive study on various geometries.

**Weaknesses:**

-  The take-away messages from this paper are too general, not specific, and really useful. As the paper mentions, in many cases, hyperbolic geometry yields the best results, there are other geometries worth consideration, e.g., Solv. Because hyperbolic geometry was studied in the previous work, the add-on Thurston geometries used in this paper cannot yield better results than hyperbolic geometry, and the embedding method used in this paper is not innovative, it is hard for me to see the scientific contributions of the paper.

-  This paper is possibly more suitable for a journal than ICLR which requires more contributions on machine/deep and representation learning aspect. Moreover, it would be more informative and useful, if the paper comes up with the concrete conclusions regarding what geometries are more suitable for what kinds of connectomes.

-  The background of hyperbolic geometry has some oversights. For instance, $g^{-}(x,y) = x_1y_1+...+x_ny_n - x_{n+1}y_{n+1}$. Moreover, it is unclear what hyperbolic model the paper talks about (i.e., Lorentz, Klein, or Poincare model).

**Questions:**

- Do you have any conclusions of what geometry should be used for what kinds of connectomes?

- For Solv, why do you need to approximate $d(a,b) = d(a,a_1)+ d(a_1, a_2) +... + d(a_k,b)$? What is $d(a_1, a_2)$ in this formulation?

---

### Official Review · Reviewer_LugH · 2023-11-07

**Soundness:** 3 good
**Presentation:** 3 good
**Contribution:** 3 good
**Rating:** 6
**Confidence:** 3

**Summary:**

This paper proposes a new embedding method for connectomes based on Simulated Annealing, which allows embed connectomes to Thurston geometries (Euclidean, spherical, hyperbolic, Solv, Nil, and other product geometries). The proposed method introduces new possibilities in modeling connectomes and is more robust than SOTA, which is crucial. Their experiments demonstrate that the proposed algorithm performs better and finds better embeddings than the SOTA. One of the key findings of this study is that the 3-dimensional hyperbolic geometry produces the best outcomes, while Solv performs as the next best alternative to embed connectomes.

**Strengths:**

1. The paper shows useful results on how Thurston geometries could be helpful in embedding connectomes.
2. The paper is well-written and structured.
3. Overall, this is an essential and comprehensive study with a reasonable amount of experiments that provide very interesting theoretical results. The authors support their theory with empirical results.

**Weaknesses:**

1. Some tables, such as Table 2, lack readability. Including indicators like up or down arrows alongside measurements such as NLL and MAP and highlighting the best outcomes will provide readers with a clear indication of value trends.
2. Some notations and abbreviations need more explanations. It's good to have consistency throughout the paper (while I did not go into the details of all the proofs, the overall sketch and techniques seem correct).
3. At some point, the authors mentioned when it comes to Euclidean geometry, the results are inconsistent. For human connectomes, E^3 outperforms other geometries. What are the possible reasons for such behaviors?

**Questions:**

1. What is the HRG model on page 4, 5th line of 2nd Paragraph? Missing references here.
2. Is there an ablation study on picking M=2000 points for most experiments?
3. Would it be possible to visualize connectomes in 3-dimensional geometries to see the algorithm's performance visually? (There are opensource libraries that could support this, e.g., geomstats)

---

### Author Response · Authors · 2023-11-23

We thank all the reviewers for their careful reading of the paper and helpful comments. We will take these comments into account when improving the paper.

Many reviewers mention things not included in the paper. Unfortunately we could not provide a detailed descriptions of geometries used, or the connectome datasets, due to space limits. These issues were described well in our references. Our dataset is the same as used in Allard & Serrano (2020), limited to the subset that is publicly available. Instead, we have focused more on extensive statistical analysis and testing, which we believe is lacking in most papers in the field.

We agree that visualizations of our embeddings would be very helpful. In particular, such a visual analysis could also help us to determine the reasons why particular geometries are better or worse for each connectome. The framework we are using (supplemental material) does have visualization capabilities. Unfortunately, we focus on 3D embeddings, which are difficult to render in paper, and again, this uses up a space, which is limited. We will consider this addition for the future version of the paper.

Answers to questions:
- HRG is the Hyperbolic Random Graph model, explained in Section 2. We will explain this acronym, thanks for catching this! In this model, the edges are indeed assumed to be independent (once the points in the hyperbolic disk are assigned).
- We are not sure how ablation study would be helpful for determining M. Our M=20000 is large enough to improve on the state-of-the-art methods in most cases. We also include an analysis comparing M=20000 to M=100000.
- We have indeed given an incorrect formula for the Minkowski inner product. Thanks for catching this! We do not understand the comment about "it is unclear what hyperbolic model the paper talks about", in all places where the model is relevant, we state the model used (Minkowski hyperboloid model to explain the basics, Poincare disk for visualization, horocyclic coordinate system to explain Solv); in the core paper, it does not matter which model is used underneath (we only care about distances between points; in some applications, the model may matter due to numerical precision issues, but our disks are small enough that these issues do not matter). (The Minkowski hyperboloid model is sometimes called the Lorentz model in ML, but this does not seem to be a popular name in general, e.g. Wikipedia does not include that name.)
- d(a1,a2) is the geometric distance between point a1 and a2. We need to approximate like that because we have found out that the method used in (Kopczyński & Celińska-Kopczyńska, 2022) does not yield usable results once the two points are too far away.

Other questions of Reviewer Qkfi: (e) in our experiments, we use Dijkstra's search just once, to construct a MxM array, which is then used for all iterations and all connectomes. This computation takes noticeable time, but it is a negligible part of the whole setup. In the last line of Page 8 we mention that the embedding time is 220 seconds per run on Mouse3 in H3 (other times can be found in the supplemental material). The dominating part is computing the new loglikelihood after the change, which is done in time O(n) where n is the number of the nodes; this is done in every iteration, so computational complexity is O(n*N_s). In our experiments, the number of iterations N_s is also linearly dependent on n. (f) we use the whole dataset to compute embeddings.

We mention "universal cover" / $\text{SL}(2,\mathcal{R}) because this name is used in some sources (e.g. Thurston 1982), so we include it so readers seeing that name in other sources know what we are talking about ("also known as"). This name is not intuitive, and explaining it would not fit the scope of the paper. Twisted product of H2xR works just like Nil (twisted product of E2xR) but the base is the hyperbolic plane.

Reviewer 7zXB: Our assumption is that the geometry of the network is different from the geometry of the Euclidean space that the connectome physically is in. This is the topic of Allard & Serrano (2020) where comparisons to physical embeddings are included, our purpose it to determine what the best network geometry it is. The paper cited by the reviewer seems to only study the physical geometry, and also we see no mention of Voronoi tessellations in it.

---

### Meta-Review · Area_Chair_GDzm · 2023-12-08

**Metareview:**

This paper studies the use of Thurston geometries to embed 2D surfaces gracefully. The authors consider a range of geometries, including isotropic geometries (such as spherical, hyperbolic, and Euclidean), product geometries (e.g., $\mathbb{H}^2 \times \mathbb{R}$), twisted product geometries, with a particular focus on Solv geometry. The embedding process is formulated as a maximum likelihood problem and approached by simulated annealing.

The paper received mixed reviews, with two rejections and two borderline acceptances from four experts. A common concern during the reviewer discussions was the lack of explanation of the unique advantages of Solv geometry, particularly in comparison to other widely-used geometries, leaving it unclear what distinctive benefits Solv geometry offers that others do not. This lack of insight is the primary concern among the reviewers. Additionally, the paper's algorithmic portion was found to be difficult to follow, with essential details and justifications either missing or not clearly articulated.

Acknowledging the merits of the work, the AC concludes that, unfortunately, the paper is not yet ready for acceptance at ICLR.

**Justification For Why Not Higher Score:**

This paper is reviewed by reviewers with solid backgrounds (Trung and Niharika) and experts in geometry (Himashi has done work on hyperbolic spaces). Both Trung and Niharika gave the paper a reject. I also read the paper and had a lot of difficulty understanding what exactly is done and why the chosen steps are optimal. As such, I firmly think that this paper cannot be accepted in its current form.

Please also note that the method is limited to mappings 2D surfaces and cannot be used for higher dimensional embeddings

**Justification For Why Not Lower Score:**

NA

---

### Decision · Program_Chairs · 2024-01-16

Reject